# Vicarious Radiometric Calibration of the Multispectral Imager Onboard SDGSAT-1 over the Dunhuang Calibration Site, China

Zhenzhen Cui [1,2], Chao Ma [1] , Hao Zhang [2,*] , Yonghong Hu [3,4] , Lin Yan [3,4], Changyong Dou [3,4] and Xiao-Ming Li [3,5]

1   School of Surveying and Land Information Engineering, Henan Polytechnic University, Jiaozuo 454003, China; 211804010010@home.hpu.edu.cn (Z.C.); mac@hpu.edu.cn (C.M.)
2   Airborne Remote Sensing Center, Aerospace Information Research Institute, Chinese Academy of Sciences, Beijing 100094, China
3   International Research Center of Big Data for Sustainable Development Goals, Beijing 100094, China; huyonghong@aircas.ac.cn (Y.H.); yanlin@aircas.ac.cn (L.Y.); doucy@aircas.ac.cn (C.D.); lixiaoming@aircas.ac.cn (X.-M.L.)
4   Key Laboratory of Digital Earth Science, Aerospace Information Research Institute, Chinese Academy of Sciences, Beijing 100094, China
5   Hainan Aerospace Information Research Institute, Wenchang 571399, China
*   Correspondence: haozhang@aircas.ac.cn

**Abstract:** The multispectral imager (MII), onboard the Sustainable Development Science Satellite 1 (SDGSAT-1), performs detailed terrestrial change detection and coastal monitoring. SDGSAT-1 was launched at 2:19 UTC on 5 November 2021, as the world's first Earth science satellite to serve the United Nations 2030 Sustainable Development Agenda. A vicarious radiometric calibration experiment was conducted at the Dunhuang calibration site (Gobi Desert, China) on 14 December 2021. In-situ measurements of ground reflectance, aerosol optical depth (AOD), total columnar water vapor, radiosonde data, and diffuse-to-global irradiance (DG) ratio were performed to predict the top-of-atmosphere radiance by the reflectance-, irradiance-, and improved irradiance-based methods using the moderate resolution atmospheric transmission model. The MII calibration coefficients were calculated by dividing the top-of-atmosphere radiance by the average digital number value of the image. The radiometric calibration coefficients calculated by the three calibration methods were reliable (average relative differences: 2.20% (reflectance-based vs. irradiance-based method) and 1.43% (reflectance-based vs. improved irradiance-based method)). The total calibration uncertainties of the reflectance-, irradiance-, and improved irradiance-based methods were 2.77–5.23%, 3.62–5.79%, and 3.50–5.23%, respectively. The extra DG ratio measurements in the latter two methods did not improve the calibration accuracy for AODs ≤ 0.1. The calibrated MII images were verified using Landsat-8 Operational Land Imager (OLI) and Sentinel-2A MultiSpectral Instrument (MSI) images. The retrieved ground reflectances of the MII over different surface types were cross-compared with those of OLI and MSI using the FAST Line-of-sight Atmospheric Analysis of Hypercubes software. The MII retrievals differed by <0.0075 (7.13%) from OLI retrievals and <0.0084 (7.47%) from MSI retrievals for calibration coefficients from the reflectance-based method; <0.0089 (7.57%) from OLI retrievals and <0.0111 (8.65%) from MSI retrievals for the irradiance-based method; and <0.0082 (7.33%) from OLI retrievals and <0.0101 (8.59%) from MSI retrievals for the improved irradiance-based method. Thus, our findings support the application of SDGSAT-1 data.

**Keywords:** vicarious calibration; reflectance-based method; irradiance-based method; Dunhuang site; SDGSAT-1





## 1. Introduction

The Sustainable Development Science Satellite 1 (SDGSAT-1) is the world's first Earth science satellite specifically dedicated to collecting data related to the Sustainable Development Goals (SDGs) by 2030 [1]. It is also the first Earth science satellite of the Chinese

Academy of Sciences (CAS). The satellite was developed by the Earth Big Data Science and Engineering Project of the CAS and is the first satellite planned by the International Research Center of Big Data for Sustainable Development Goals. It carries three main payloads: the multispectral imager (MII), the glimmer imager (GLI), and the thermal infrared spectrometer (TIS), which are aimed at meeting the needs for monitoring, evaluation, and research of the global SDGs. Through the joint and continuous observation by the three imagers, SDGSAT-1 provides ample data support for the study of indicators representing human–nature interactions, the fine characterization of "traces of human activities", and the implementation of the global SDGs. SDGSAT-1 is expected to lead in reducing the global imbalance in sustainable development and the digital divide between regions. Most importantly, all SDGSAT-1 data are shared globally and freely, and they can be downloaded from the SDGSAT-1 Data Open System (http://124.16.184.48:6008, accessed on 9 June 2022).

Radiometric calibration is an essential assurance for the quantitative application of remote sensing data. Although the performance specifications of the satellite sensors were precisely evaluated in the laboratory before launch, the spectral and radiometric performance of the sensors are affected by post-launch changes, including space environment effects and the radiometric degradation of calibration equipment and the imaging sensor. To guarantee the MII's radiometric precision, vicarious radiometric calibration in the post-launch stage is required [2]. Vicarious radiometric calibration is most widely performed using in-situ experiments [3]. The reflectance-, irradiance-, and radiance-based methods, which were first proposed and applied in radiometric calibration tests conducted by the Remote Sensing Group of the University of Arizona, are three representative in-situ vicarious radiometric calibration methods [4]. According to data published by the United States, France, and other countries, the accuracy of absolute radiometric calibration using the current vicarious radiometric calibration methods in the visible and near-infrared bands is 3–5% [5]. The reflectance-based method depends on ground-based measurements, including ground reflectance measurements and atmospheric parameters. In addition, the top-of-atmosphere (TOA) radiance is obtained by entering these parameters into a radiative transfer model. The irradiance-based method is similar to the reflectance-based method; the main difference is the addition of the diffuse-to-global irradiance (DG) ratio measurement to reduce errors introduced by aerosol model assumption. The radiance-based method requires a strictly calibrated radiometer onboard the airplane and an uncalibrated sensor to obtain the radiance of the target under nearly simultaneous and consistent observation conditions. The measured radiance is corrected according to the atmospheric influence between the radiometer and the sensor.

Since the 1980s, numerous satellite radiometric calibration sites have been constructed in many different nations, consisting of White Sands Missile Range [3], Railroad Valley Playa [6], Lunar Lake Playa [7], Rogers Dry Lake [8], and Ivanpah Playa [9] in the United States; La Crau [10] in France; Newell Country [11] in Canada; Tinga Tingana [12], Uardry [13], and Lake Frome [14] in Australia; and Dunhuang [15], Qinghai Lake [16], and Baotou [17] in China. These calibration sites have provided solid support for the radiometric calibration of numerous optical sensors, including the thermal mapper [18,19], the enhanced thermal mapper plus [20,21], and the operational land imager (OLI) [22–24] onboard Landsat; the high-resolution visible onboard SPOT [25]; the moderate resolution imaging spectroradiometer onboard Terra and Aqua [26,27]; the multispectral sensor onboard the ZY satellite [28]; and the hyperspectral imager onboard the HJ-1A satellite [29]. Calibration results have shown that the radiance-based method has the maximum radiometric calibration accuracy among the three methods and that the higher the airplane's altitude, the higher the accuracy of this method. Nevertheless, the high cost of manpower and material resources, the strictly simultaneous measurements between the airplane and the satellite, the high accuracy of the radiometer, and the low success rate greatly limit the application of the radiance-based method. The irradiance-based method has lower accuracy than the radiance-based method but is superior to the reflectance-based method, particularly under poor atmospheric conditions with large aerosol optical depth (AOD). To

solve the problem of low calibration accuracy under unstable atmospheric conditions, novel methods have been put forward in the past years, including the supervised vicarious [30] and the improved irradiance-based [31] methods, which have been effectively used for the vicarious radiometric calibration of an unmanned aerial vehicle hyperspectral sensor [31], airborne hyperspectral sensors [32], and the SPARK-01/02 satellites [33].

To further validate these methods, an experiment was performed at the Dunhuang calibration site in China on 14 December 2021, to calibrate the MII. This was the first in-situ calibration experiment of the MII and is expected to provide reliable radiometric calibration coefficients. Calibration uncertainty analysis is discussed in detail in this article. Two reference sensors, i.e., the OLI onboard Landsat-8 and the MSI onboard Sentinel-2A, were selected to cross-validate the results of the vicarious radiometric calibration.

The remainder of this article is divided into the following sections: the data, measurements, and vicarious radiometric calibration methods employed in this investigation are thoroughly described in Section 2. The MII calibration results are listed in Section 3. Finally, in Section 5, we summarize the key findings of our research.

## 2. Materials and Methods

### 2.1. Materials

#### 2.1.1. Overview of SDGSAT-1

SDGSAT-1, developed jointly by the Aerospace Information Research Institute, Changchun Institute of Optics, Fine Mechanics and Physics, the Shanghai Institute of Technical Physics, and the Innovation Academy for Microsatellites (CAS, China), was successfully launched by the CZ-6 rocket at the Taiyuan Satellite Launch Center at 2:19 UTC on 5 November 2021. SDGSAT-1 carries three sensors (MII, GLI, and TIS) to observe the Earth's surface continuously during the day and night. The MII is anticipated to deliver multispectral data over inland water bodies and terrestrial surfaces. The GLI and TIS were designed to acquire global night data and thermal infrared data, respectively. Table 1 lists the specific parameters of SDGSAT-1. For detailed information on SDGSAT-1, please visit http://www.sdgsat.ac.cn/ (accessed on 29 July 2022).

**Table 1.** Major technical parameters of SDGSAT-1.

| SDGSAT-1 | Parameters |
|---|---|
| Payloads | MII, GLI, TIS |
| Spectral bands | MII: 7 bands ranging from 374 to 911 nm<br>GLI: 4 bands ranging from 424 to 910 nm<br>TIS: 3 bands ranging from 8 to 12.5 μm |
| Swath | 300 km |
| Orbit | Type: sun-synchronous orbit<br>Altitude: 505 km<br>Inclination: 97.5° |
| Spatial resolution | MII: 10 m @ 505 km<br>GLI: 10 m @ 505 km (PAN), 40 m @ 505 km (B, G, R)<br>TIS: 30 m @ 505 km |
| Revisit period | 11 days |

#### 2.1.2. SDGSAT-1 MII

The MII instrument, designed by the Changchun Institute of Optics, Fine Mechanics and Physics and the Aerospace Information Research Institute CAS, comprises two cameras, A and B, with eight detectors per camera. Seven spectral bands are set in the visible and near-infrared bands: B1 (coastal/aerosol 1), B2 (coastal/aerosol 2), B3 (blue), B4 (green), B5 (red), B6 (red edge), and B7 (near infrared). The MII instrument was designed to have a large swath width of 300 km at a nominal altitude of 505 km. The spatial resolution is approximately 10 m in all seven bands, providing high-resolution data support for SDGs 2, 6, 11, 13, 14, and 15 [34]. Figure 1 depicts the relative spectral response function of the SDGSAT-1 MII, while Table 2 displays the major technical parameters.

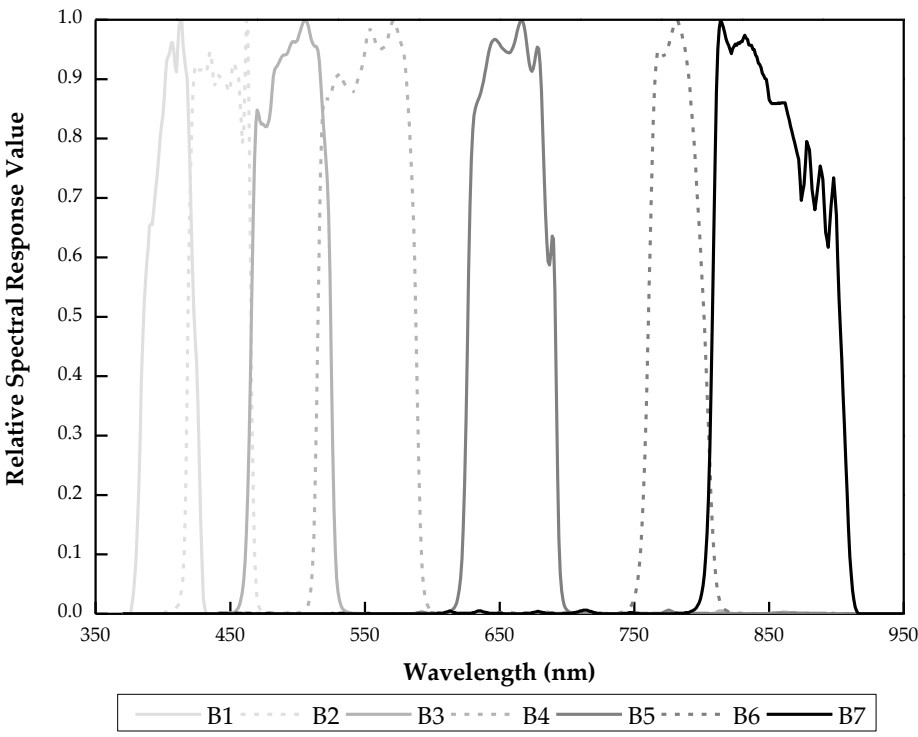

**Figure 1.** Relative spectral response function of the SDGSAT-1 MII.

**Table 2.** Major technical parameters of the SDGSAT-1 MII.

| SDGSAT-1 MII | Description | Technical Parameters |
|---|---|---|
| | B1 | 374–427 nm |
| | B2 | 410–467 nm |
| | B3 | 457–529 nm |
| Spectral range | B4 | 510–597 nm |
| | B5 | 618–696 nm |
| | B6 | 744–813 nm |
| | B7 | 798–911 nm |
| Spatial resolution | B1–B7 | 10 m @ 505 km |
| Signal-to-noise ratio | B1 | ≥130 |
| | B2–B7 | ≥150 |
| Swath width | B1–B7 | 300 km |
| Modulation transfer | Static | ≥0.23 |
| function | Dynamic | ≥0.10 |
| Dynamic range | B1–B7 | ≥60 dB |
| Digitalizing bits | B1–B7 | 12 bits |

### 2.1.3. Calibration Site and Image Data Acquisition

The Dunhuang calibration site, established at the end of the 20th century, is one of the China Radiometric Calibration Sites for the vicarious radiometric calibration of satellite sensors, as well as an internationally recognized radiometric calibration site with flat terrain, uniform surface, stable and measurable ground objects, and good directional characteristics [2]. The Dunhuang calibration site is located in northwest China on the regeneration alluvial fan of the Danghe River more than 20 km northwest of Dunhuang City. The calibration site has a total area of approximately 30 km × 30 km; its geographical coordinates are 40.04–40.28°N and 94.17–94.5°E; and its altitude is 1160 m. As shown in Figure 2, the region utilized for vicarious radiometric calibration tests of high- and medium-spatial resolution sensors is roughly 500 m × 500 m and is located in the middle of the alluvial fan. The entire alluvial fan is fairly flat, and the surface is mainly evenly distributed gravel with diameters of 0.2–8 cm [35]. The ground surface of the Dunhuang site

is relatively flat, with low vegetation coverage and high spectral stability. The atmosphere is dry and clean, with a low AOD (in the absence of sandstorms, the local AOD is 0.1–0.2) and low total columnar water vapor (CWV). The sunshine duration is long. Therefore, the Dunhuang site is conducive for calibration experiments. Vicarious radiometric calibration of multiple domestic satellites, including BJ-1 [36], FY-1C [37], HJ-1A [29], CBERS-02 [38], and SPARK-01/02 [33], was carried out at the Dunhuang calibration site. The SDGSAT-1 MII data over the Dunhuang calibration site were acquired at 03:45:17 UTC on 14 December 2021. Table 3 lists the observation geometries on the SDGSAT-1 overpass date.

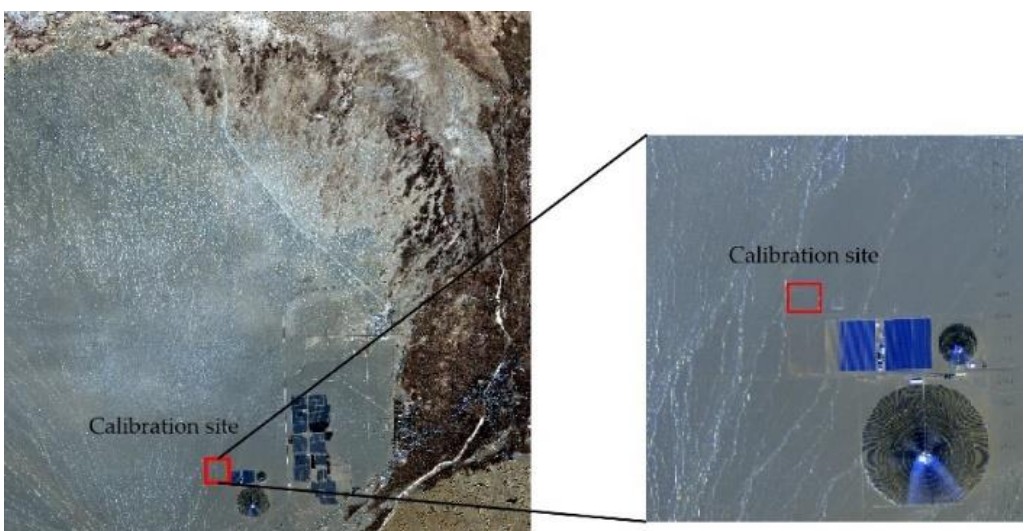

**Figure 2.** Sentinel-2A/MSI image of the Dunhuang calibration site acquired on 17 December 2021.

**Table 3.** Information on the capture of SDGSAT-1 MII image at the Dunhuang calibration site.

| Date | Overpass Time (UTC) | Solar Zenith (°) | Solar Azimuth (°) | View Zenith (°) | View Azimuth (°) |
|---|---|---|---|---|---|
| 14 December 2021 | 03:45:17 | 68.5554 | 152.2536 | 18.1581 | 304.6388 |

2.1.4. Simultaneous Measurement of Ground Reflectance

The internationally popular FieldSpec-4 ASD spectroradiometer (ASD Inc., Longmont, CO, USA), with a wavelength range of 350–2500 nm, was used for simultaneous ground reflectance measurements. The calibration site was a 500 m × 500 m square region, with SDGSAT-1 MII coverage of approximately 50 cross-track pixels and 50 along-track pixels. Ground reflectance measurements were performed in the square region (Figure 2) from 10:30 to 12:30 UTC + 8. Ground reflectance was measured along a designed route, as shown in Figure 3b. Adjacent measurement points were ~20 m apart. Five measurements were taken around each point, and five spectra were collected for each measurement. The total measured data exceeded 400 groups, with nearly 2000 spectral data points of the Dunhuang calibration site. After removing abnormal and erroneous measurements, all valid measurements were averaged to represent the final reflectance of the Dunhuang calibration site (Figure 4).

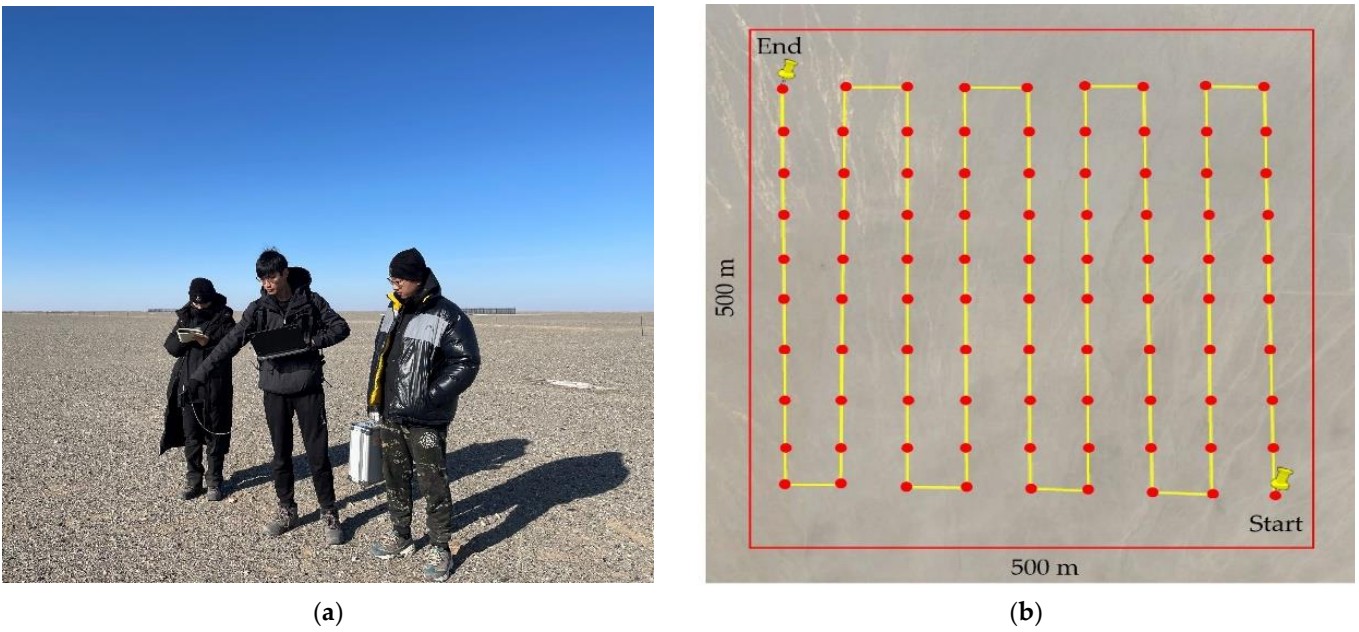

(**a**)          (**b**)

**Figure 3.** (**a**) Photograph of the in-situ ground reflectance measurement at Dunhuang calibration site. (**b**) Scheme of the ground surface measurement route.

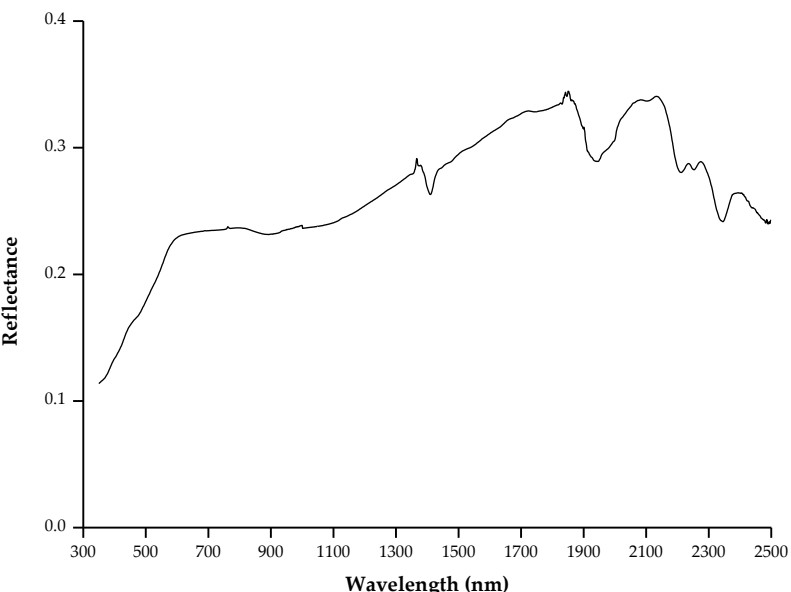

**Figure 4.** Simultaneous ground reflectance measurements performed over the Dunhuang calibration site on 14 December 2021, using FieldSpec-4 ASD spectroradiometer.

2.1.5. Simultaneous Measurement of Atmospheric Parameters

Simultaneous atmospheric parameters (AOD, CWV, the DG ratio, and the atmospheric vertical profile) were acquired on 14 December 2021. Total AOD and CWV were measured using an automated CIMEL CE318 sun photometer (Cimel Electronique, Paris, France) and retrieved using an improved Beer–Lambert–Bouguer method [39] and a four-parameter method [40]. The AOD values at the 440 nm and 670 nm channels were employed to derive the AOD at the 550 nm channel via logarithmic interpolation, using Equation (1). Figure 5 shows the 550 nm AOD and CWV on the SDGSAT-1 overpass date.

$$\tau_\alpha(\lambda) = \beta \cdot \lambda^{-\alpha} \tag{1}$$

$$\alpha = -\frac{\ln(\text{AOD}(\lambda_m))/\text{AOD}(\lambda_n))}{\ln(\lambda_m/\lambda_n)} \tag{2}$$

$$\beta = \frac{\text{AOD}(\lambda_m)}{\lambda_m{}^{-\alpha}} = \frac{\text{AOD}(\lambda_n)}{\lambda_n{}^{-\alpha}} \tag{3}$$

where $\tau_\alpha(\lambda)$ is the AOD at wavelength $\lambda$ (in μm), $\alpha$ is the Ångstrom wavelength exponent, and $\beta$ is the atmospheric turbidity coefficient.

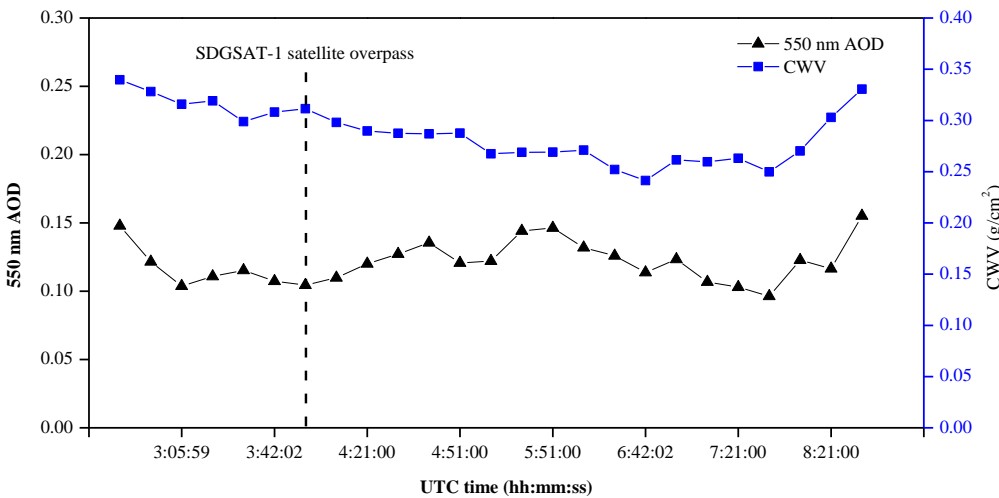

**Figure 5.** The 550 nm AOD and CWV retrieved from the CE318 sun photometer on 14 December 2021.

Data retrieved from the CE318 sun photometer within 5 min before and after the overpass time were averaged to obtain the 550 nm AOD and CWV at the overpass time. A level-3 ozone data product (https://acdisc.gsfc.nasa.gov/data/Aura_OMI_Level3/OMDOAO3 e.003/2021/, accessed on 21 December 2021) retrieved from the ozone monitoring instrument onboard the Aura satellite was used as the columnar ozone content at the overpass time. Table 4 presents the simultaneous measurements of atmospheric parameters at the SDGSAT-1 overpass time.

**Table 4.** Simultaneous measurements of atmospheric parameters.

| Atmospheric Parameters | Simultaneous Measurements at Overpass Time |
| --- | --- |
| 550 nm AOD | 0.1045 |
| CWV | 0.3114 g/cm$^2$ |
| Columnar ozone content | 301.6 DU |

An automated spectral radiometer was used to measure the DG ratio within the Dunhuang calibration site. The DG ratio was recorded at 6 min intervals during the daytime. Figure 6a,b show the DG ratios at 550 nm throughout the day of the SDGSAT-1 overpass and the entire spectrum at the SDGSAT-1 overpass time, respectively. Abnormal data were eliminated to derive a smoothed DG ratio curve, which indicated a stable atmospheric condition most of the time, with some exceptions (e.g., at around 2:17 and 7:48 UTC), on 14 December 2021.

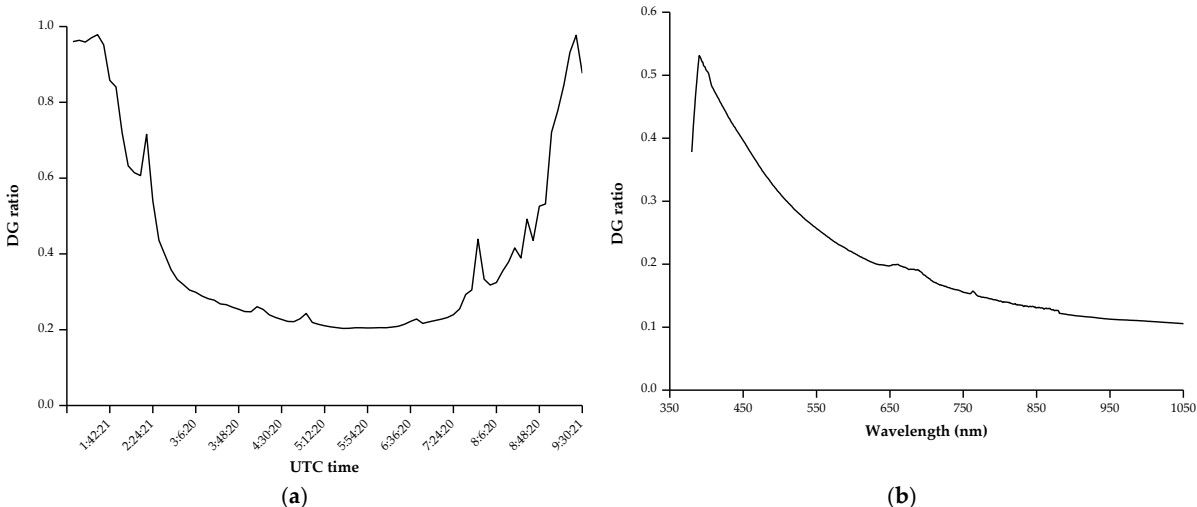

**Figure 6.** Measured DG ratios (**a**) at 550 nm on the date of the SDGSAT-1 overpass and (**b**) the entire spectrum at the time of the SDGSAT-1 overpass.

A radiosonde balloon was used to obtain the atmospheric vertical profiles of pressure, temperature, and humidity on the SDGSAT-1 overpass date. The balloon was released from the Dunhuang National Reference Climate Station at 07:15 a.m. (Beijing time) on 14 December 2021. Figure 7 shows the measured atmospheric vertical profiles of pressure, temperature, and relative humidity, which varied with altitude, on 14 December 2021.

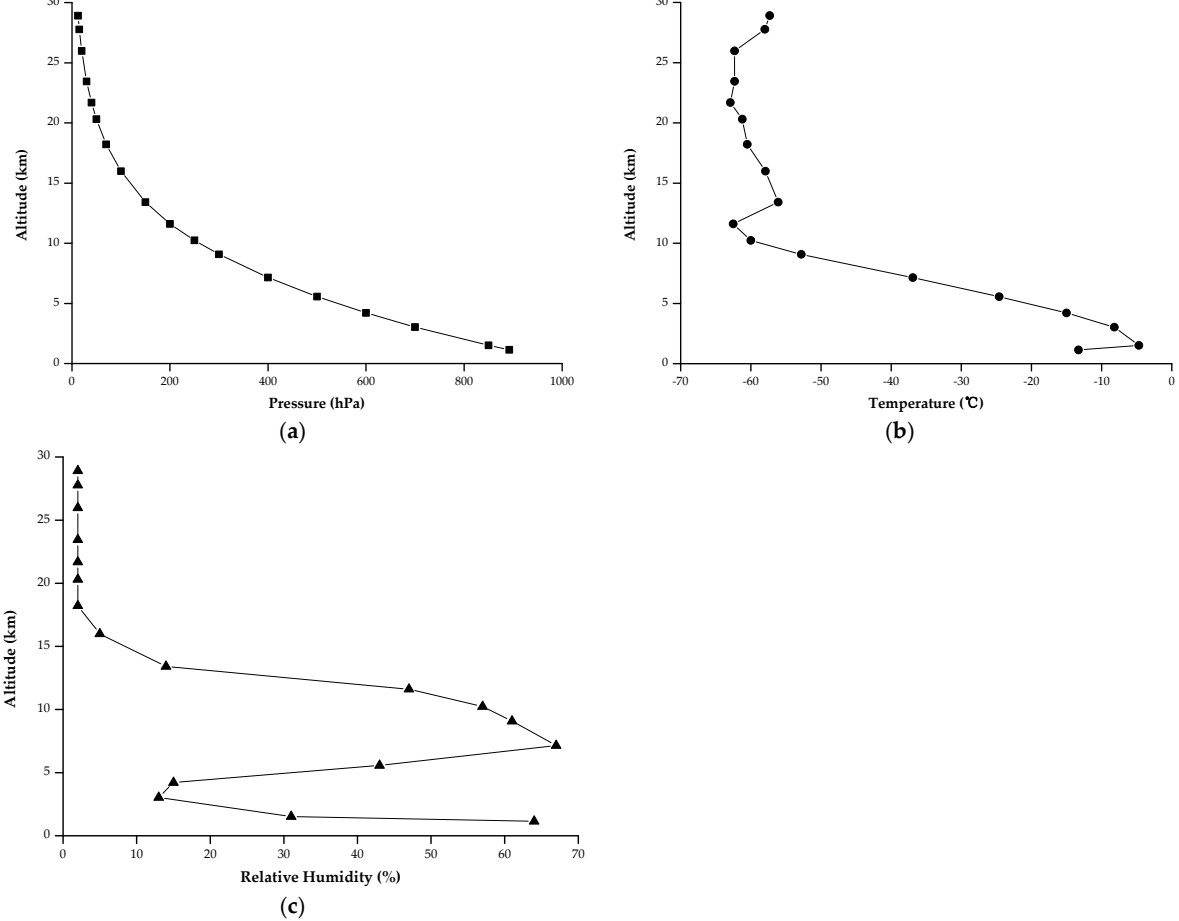

**Figure 7.** Atmospheric vertical profiles of (**a**) pressure, (**b**) temperature, and (**c**) relative humidity obtained by the radiosonde balloon released on 14 December 2021.

*2.2. Methods*

2.2.1. Radiometric Calibration Method

The widely used reflectance- [4], irradiance- [18], and improved irradiance-based [31,33] methods were adopted to predict the TOA radiance. All three methods rely on the accurate simultaneous measurements of the ground reflectance and atmospheric parameters. The reflectance-based method for radiometric calibration is generally used in clear and cloudless weather conditions. It requires fewer surface observations and less labour input than the other methods and is easy to operate. However, the reflectance-based method requires high visibility and a stable atmosphere to diminish the calibration uncertainty caused by aerosol model assumptions. The total calibration uncertainty significantly increases in the case of a high AOD [33]. The irradiance-based method is an enhanced reflectance-based method that requires simultaneous measurements of the ground reflectance and the DG ratio. The latter parameter is utilized to replace the aerosol model assumption in the reflectance-based method. The aerosol model assumption in the irradiance-based method only affects path radiance and the hemisphere albedo in Equation (7), while large uncertainties in the upward and downward scattering transmittance are minimized by using the measured DG ratio. The reflectance-based method has a total uncertainty of 4.9% [41], and the simultaneous ground reflectance in in-situ measurements and the aerosol model assumption in the radiative transfer calculation are its main sources of uncertainty. The irradiance-based method generally outperforms the reflectance-based method with an uncertainty of 3.5% [41], and the DG ratio in in-situ measurements is its main source of uncertainty. However, in the case of sunny weather and a low AOD, the calibration accuracy of the reflectance-based method is comparable to that of the irradiance-based method [31]. The vicarious radiometric calibration equations for the reflectance- and irradiance-based methods used for the satellite sensors are Equations (4) and (7), respectively. The TOA radiance can be converted using Equation (8) [33]. Equations (9) and (10) were used to determine the spectral radiometric calibration coefficients on the assumption that the sensor response was linear.

$$\rho^*(\theta_s, \theta_v, \varphi_{v-s}) = \rho_a(\theta_s, \theta_v, \varphi_{v-s}) + \frac{\rho_t}{1 - \rho_t \times s} \times T(\theta_s) \times T(\theta_v) \tag{4}$$

$$T(\theta_s) = T_{dir}(\theta_s) + T_{dif}(\theta_s) = (1 - \rho_t \times s) \times \frac{e^{-\tau/\mu_s}}{1 - \alpha_s} \tag{5}$$

$$T(\theta_v) = T_{dir}(\theta_v) + T_{dif}(\theta_v) = (1 - \rho_t \times s) \times \frac{e^{-\tau/\mu_v}}{1 - \alpha_v} \tag{6}$$

$$\rho^*(\theta_s, \theta_v, \varphi_{v-s}) = \rho_a(\theta_s, \theta_v, \varphi_{v-s}) + \frac{e^{-\tau/\mu_s}}{1 - \alpha_s} \times \rho_t \times (1 - \rho_t \times s) \times \frac{e^{-\tau/\mu_v}}{1 - \alpha_v} \tag{7}$$

$$L = \rho^* \times \cos(\theta_s) \times E_0 / \left( d^2 \times \pi \right) \tag{8}$$

$$L_i(\lambda) = \frac{\int L(\lambda) \cdot RSR_i(\lambda) d\lambda}{\int RSR_i(\lambda) d\lambda} \tag{9}$$

$$L_i = DN_i \cdot Gain_i + Bias_i \tag{10}$$

where $\theta_s$ and $\theta_v$ are the sun zenith angle and the view zenith angle, respectively; $\varphi_{v-s}$ is known as the relative azimuth angle between the view azimuth angle and the sun azimuth angle. $\rho_t$ is the measured ground reflectance, and $\rho_a$ is the atmospheric intrinsic reflectance. $s$ is the atmospheric hemisphere reflectance. $T(\theta_s)$ and $T(\theta_v)$ are the total transmittances of both the downward direction (solar path) and the upward direction (view path), respectively; $T_{dir}(\theta_s)$ and $T_{dif}(\theta_s)$ are the direct and diffuse transmittances in the downward direction, and $T_{dir}(\theta_v)$ and $T_{dif}(\theta_v)$ are the direct and diffuse transmittances in

the upward direction. $\rho^*$ and $L$ are the TOA reflectance and TOA radiance of the surface target, respectively. $\mu_s$ and $\mu_v$ refer to the values of $\cos(\theta_s)$ and $\cos(\theta_v)$, respectively; $\alpha_s$ and $\alpha_v$ represent the DG ratios of the solar direction and viewing direction, respectively. d is the Sun–Earth distance in astronomical units and $E_0$ is the TOA solar irradiance. $L_i$ represents the TOA radiance for the $i$-th band. $RSR_i$ is the relative spectral response for the $i$-th band. $DN_i$ is the digital number (DN) derived from the L1A image for the $i$-th band, and $Gain_i$ and $Bias_i$ are the calibration coefficients for the $i$-th band.

Even though the measured DG ratio used in the irradiance-based method would greatly reduce the uncertainty error caused by the aerosol model assumption, it is difficult to directly measure the DG ratio in the observation direction ($0°$ of view zenith angle in most cases) in in-situ experiments. Therefore, it has to be extrapolated by fitting the values at different solar zenith angles. There is a linear relationship if the atmospheric conditions are stable, as shown in Equation (11) [4].

$$ln(1 - \alpha_s) = ln(1 - \rho_t s) - (1 - b)\tau m \qquad (11)$$

where $m$ is the relative optical air mass (i.e., the inverse of the cosine of solar zenith, $1/\mu_s$) and $-(1 - b)\tau$ and $ln(1 - \rho_t s)$ are the slope and intercept of the linear fitting equation, respectively.

Ideal stable atmospheric conditions are not guaranteed in in-situ experiments. Therefore, when the atmospheric condition is unstable, but DG ratio data are used for satellite vicarious radiometric calibration, only $\alpha_s$ is employed to substitute for the scattering effect in the reflectance-based method, as shown in Equation (12) [31], which is the improved irradiance-based method. This method has been used for vicarious calibration of the SPARK-01/-02 satellites [33] and an unmanned aerial vehicle hyperspectral sensor [31].

$$\rho^*(\theta_s, \theta_v, \varphi_{v-s}) = \rho_a(\theta_s, \theta_v, \varphi_{v-s}) + \frac{\rho_t \times e^{-\tau/\mu_s}}{1 - \alpha_s} \times T(\theta_v) \qquad (12)$$

The relationship between the ratio of $ln(1 - \alpha_s)$ and the relative optical air mass ($m$) at 550 nm measured on 14 December 2021 is represented by a scatter plot (Figure 8), and it revealed a nearly linear relationship of the measurements, with an $R^2$ value of 0.9965, indicating a relatively stable atmospheric condition on this date. Figure 9 shows goodness-of-fit ($R^2$) statistics for the DG ratio measurements in the entire visible to near infrared spectral range. The DG ratios in both the solar and viewing directions at the SDGSAT-1 overpass time are shown in Figure 10.

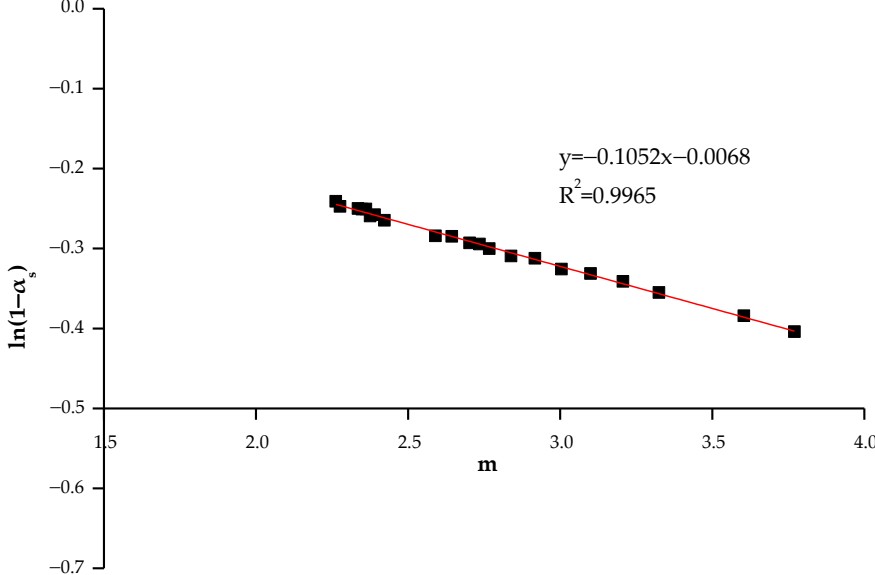

**Figure 8.** Scatter plot of $ln(1 - \alpha_s)$ versus $m$ at 550 nm.

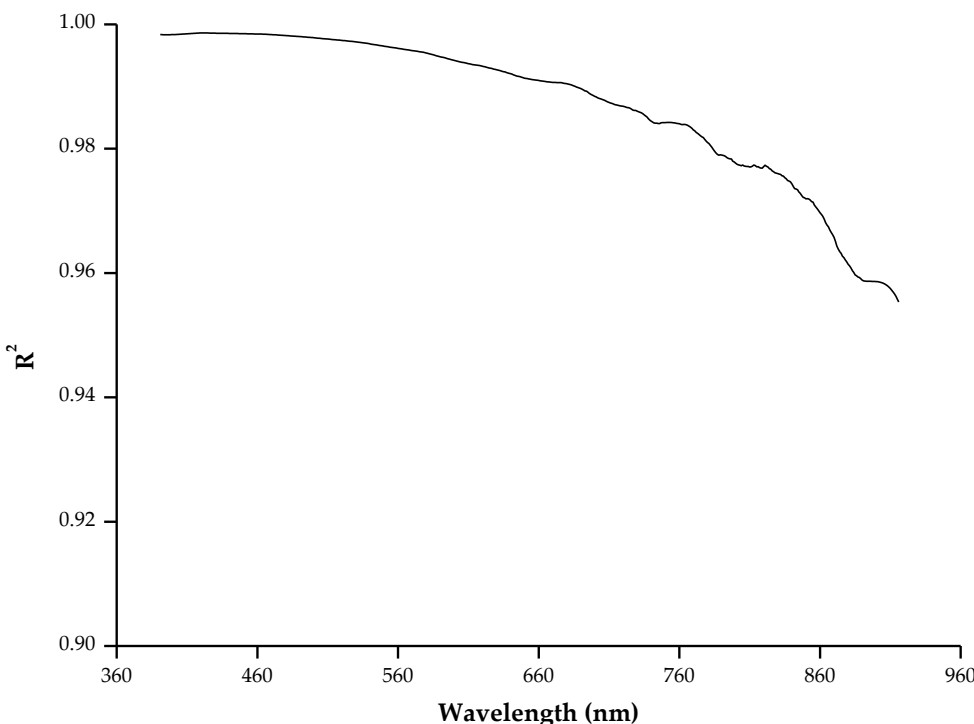

**Figure 9.** Goodness-of-fit statistics for DG ratio measurements according to Equation (11).

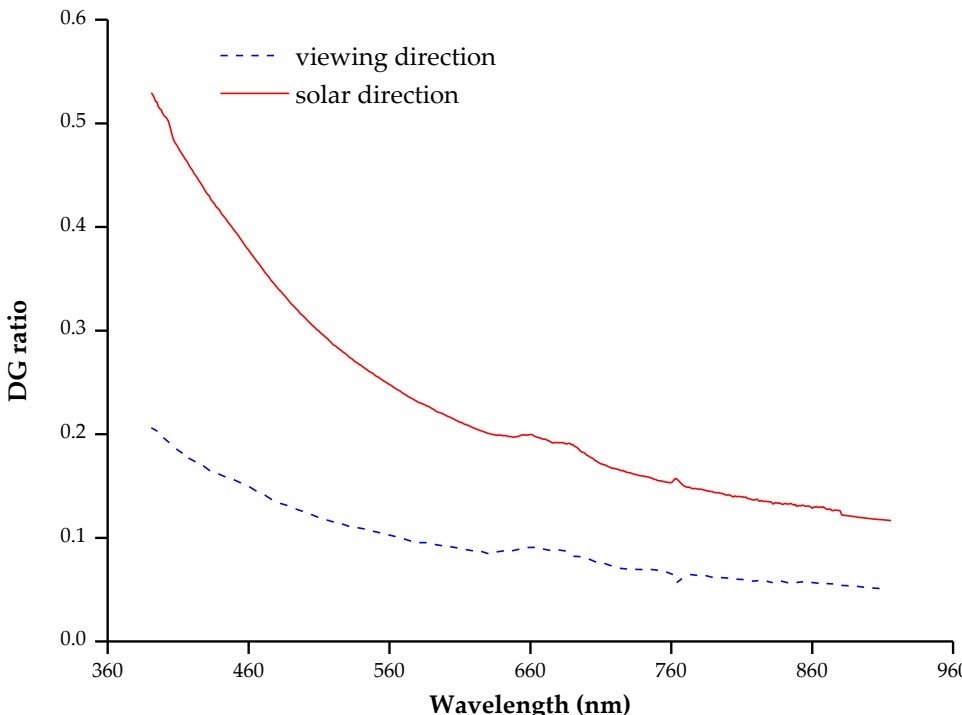

**Figure 10.** DG ratios in solar and viewing directions during the SDGSAT-1 overpass.

### 2.2.2. Radiative Transfer Calculations

The moderate resolution atmospheric transmission (MODTRAN) model is currently one of the most frequently used radiative transfer models for radiometric calibration. The spectral range of MODTRAN is 0.3–100 μm, which allows the calculation of atmospheric radiative transfer from the visible, near-infrared, mid-infrared, and far-infrared bands [42]. The measured ground reflectance, spectral response function of the SDGSAT-1 MII sensor,

550 nm AOD, CWV, ozone content, image geometric parameters, and other parameters were input into MODTRAN v.5.2.1 software. The Ångstrom index was used to constrain the band scattering characteristics of the default aerosol type in the original radiative transfer model, and then the band-equivalent TOA spectral radiance of the MII sensor was obtained. Assuming that each MII channel had a linear response and that the bias was zero because of the low dark current of the instrument (Figure 11), the TOA radiance and the average DN value of the image of each channel were substituted into Equation (13) to obtain the calibration coefficient $Gain_i$ of each channel.

$$Gain_i = \frac{L_i}{\overline{DN_i}} \tag{13}$$

where $L_i$ is the band-equivalent TOA spectral radiance ($W \cdot m^{-2} \cdot sr^{-1} \cdot \mu m^{-1}$), $\overline{DN_i}$ is the average DN of the selected area of the satellite image in band $i$, and $Gain_i$ is the radiometric calibration coefficient $Gain$ in band $i$.

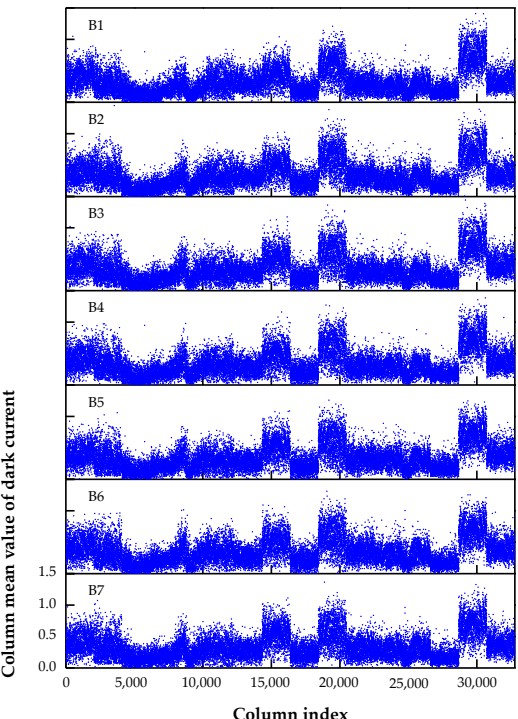

**Figure 11.** The column mean value of dark current acquired by imaging the ocean at night on 14 December 2021.

### 2.2.3. Calibration Uncertainty Estimation

The relative difference is used to represent the vicarious radiometric calibration uncertainty, and the uncertainty of each factor on the calibration coefficient $\varepsilon_i$ is calculated according to Equation (14). $L_{Original_i}$ is the reference spectral TOA radiance, and $L_{New_i}$ is the spectral TOA radiance after changing the input conditions. The final total uncertainty is expressed as the square root of the sum of the squares of each error uncertainty [31], as described by Equation (15).

$$\varepsilon_i = \left| \frac{L_{New_i} - L_{Original_i}}{L_{Original_i}} \right| \times 100\% \tag{14}$$

$$\varepsilon_{Total} = \sqrt{\varepsilon_1{}^2 + \varepsilon_2{}^2 + \varepsilon_3{}^2 + \ldots + \varepsilon_n{}^2} \tag{15}$$

### 2.2.4. Cross Validation

Given that the relative spectral response of each band differs among sensors, the ground reflectance of the "reference sensor" has to be transferred to the "target sensor" using a spectral band adjustment factor (SBAF). The divergences in the relative spectral response between various satellite sensors were computed by the SBAF using Equation (16):

$$SBAF = \frac{\hat{\rho}_{Reference}}{\hat{\rho}_{Target}} = \frac{\frac{\int \rho_{in-situ}(\lambda)RSR_{Reference}(\lambda)d\lambda}{\int RSR_{Reference}(\lambda)d\lambda}}{\frac{\int \rho_{in-situ}(\lambda)RSR_{Target}(\lambda)d\lambda}{\int RSR_{Target}(\lambda)d\lambda}} \tag{16}$$

where $\rho_{in-situ}(\lambda)$ is the per-wavelength in-situ ground reflectance at the Dunhuang calibration site, $RSR_{Reference}$ and $RSR_{Target}$ are the per-wavelength relative spectral response curves from the "reference sensor" and the "target sensor," and $\hat{\rho}_{Reference}$ and $\hat{\rho}_{Target}$ are the in-situ ground reflectance values based on the integration of the "reference sensor" and "target sensor" values. The ground reflectance of the "reference sensor" was transferred to the "target sensor" using the SBAF according to Equation (17):

$$\rho'_{Target} = \rho_{Target} \times SBAF \tag{17}$$

### 3. Results

The SDGSAT-1 TOA radiance simulated by MODTRAN 5.2.1 according to the reflectance-, irradiance-, and improved irradiance-based methods for the SDGSAT-1 MII is shown in Figure 12. Slight differences in the TOA spectral radiance were observed among the three methods.

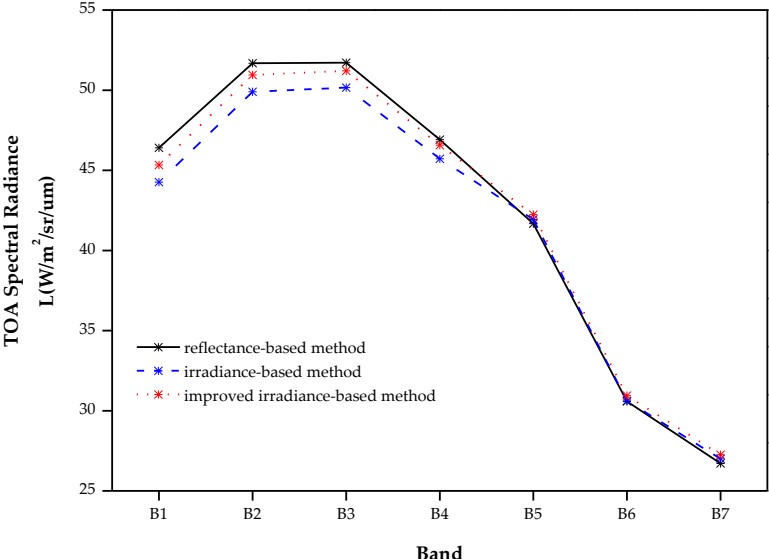

**Figure 12.** MODTRAN-simulated TOA spectral radiances for the SDGSAT-1 MII calculated by the reflectance-, irradiance-, and improved irradiance-based methods.

Table 5 lists the relative differences in TOA spectral radiances simulated by MODTRAN between the reflectance-based method and the irradiance- and improved irradiance-based methods. The relative differences between the reflectance- and irradiance-based methods varied from 0.12% to 4.61% in different bands, while differing from 0.73% to 2.31% between reflectance- and improved irradiance-based methods. In addition, the relative differences for B1 to B3 were larger than that of B4 to B7, indicating a higher radiometric calibration accuracy for B4 to B7 compared with that of B1 to B3. As expected, the strong aerosol scattering effects in the shorter spectral range cause much larger calibration uncertainty in the first three bands than in the other bands. In general, the reflectance-based

method has a similar accuracy with the irradiance- and improved irradiance-based methods when the AOD is small. However, the maximum difference between the reflectance- and irradiance-based methods can reach up to 4.61%, which was attributed either to the measurement uncertainties of the DG ratio in the solar direction or the extrapolation uncertainties in the viewing direction. Such kind of difference was seldom mentioned in the previous literature and will be discussed in later sections. As a comparison, the difference between reflectance- and improved irradiance-based methods was lower because only the DG ratio in the solar direction was used in the latter method. The average relative difference between the reflectance- and improved irradiance-based methods (1.43%) was lower than that between the reflectance- and irradiance-based methods (2.20%) because only the DG ratio in the solar direction was used in the improved irradiance-based method. In addition, the relative differences for B1 to B3 were larger than those for B4 to B7, indicating a higher radiometric calibration accuracy for the latter bands.

**Table 5.** Relative differences in TOA spectral radiances between the reflectance-based method and the irradiance- and improved irradiance-based methods.

| Band | Relative Difference between Reflectance- and Irradiance-Based Methods | Relative Difference between Reflectance- and Improved Irradiance-Based Methods |
|---|---|---|
| B1 | 4.61% | 2.31% |
| B2 | 3.44% | 1.41% |
| B3 | 3.00% | 0.99% |
| B4 | 2.54% | 0.73% |
| B5 | 0.63% | 1.35% |
| B6 | 0.12% | 1.18% |
| B7 | 1.09% | 2.04% |
| Average relative difference | 2.20% | 1.43% |

The calibration coefficients of the MII were calculated by dividing the TOA radiance by the average DN of the 40 × 40 pixels. Table 6 lists the vicarious radiometric calibration coefficients derived by the three different methods.

**Table 6.** Vicarious radiometric calibration coefficients for the SDGSAT-1 MII (unit: $W \cdot m^{-2} \cdot sr^{-1} \cdot \mu m^{-1} \cdot DN^{-1}$).

| Band | Reflectance-Based Method | Irradiance-Based Method | Improved Irradiance-Based Method |
|---|---|---|---|
| B1 | 0.051616908 | 0.049237722 | 0.050422238 |
| B2 | 0.036291910 | 0.035042783 | 0.035781749 |
| B3 | 0.023327113 | 0.022627800 | 0.023095942 |
| B4 | 0.015849453 | 0.015446672 | 0.015733488 |
| B5 | 0.016096157 | 0.016197140 | 0.016314004 |
| B6 | 0.019731394 | 0.019754163 | 0.019963674 |
| B7 | 0.013811256 | 0.013961794 | 0.014092721 |

## 4. Discussion

### 4.1. Uncertainty Analysis

The uncertainty errors associated with in-situ simultaneous measurements, data processing, and calibration method selection can be directly or indirectly attributed to the total uncertainty in the satellite sensor calibration calculation results [35,43,44]. In this section, we mainly focus on the uncertainties caused by aerosol model assumption, atmospheric model assumption, AOD retrieval, water vapor retrieval, ground reflectance measurements, viewing geometry error, radiative transfer model-inherent errors, and DG ratio measurement. The calibration uncertainty contributed by each parameter was estimated from the

difference between the TOA radiance predicted when errors were added to each parameter and the original measurement (Equation (14)).

### 4.1.1. Uncertainty Analysis of Aerosol Model Assumptions

A previous study revealed that the natural aerosol type at the Dunhuang calibration site is similar to the rural and desert types in MODTRAN [33]. To estimate the uncertainty caused by different aerosol types, desert, urban, and maritime aerosol types were input in sequence while other parameters remained unchanged during the MODTRAN calculation, and the TOA spectral radiance of the output for the three aerosol types was compared with that of the rural type. The Ångstrom exponent coefficients determined by multiband aerosol optical thickness derived from CE318 measurement data can be used as input to constrain the band scattering characteristics of the default aerosol type in the original radiative transfer model to improve the calculation accuracy of aerosol scattering in each band [41]. The Ångstrom index of SDGSAT-1 is 0.7938. The relative differences in the TOA spectral radiance for different aerosol types for the reflectance-, irradiance-, and improved irradiance-based methods are exhibited in Figure 13. The average relative differences are provided in Table 7.

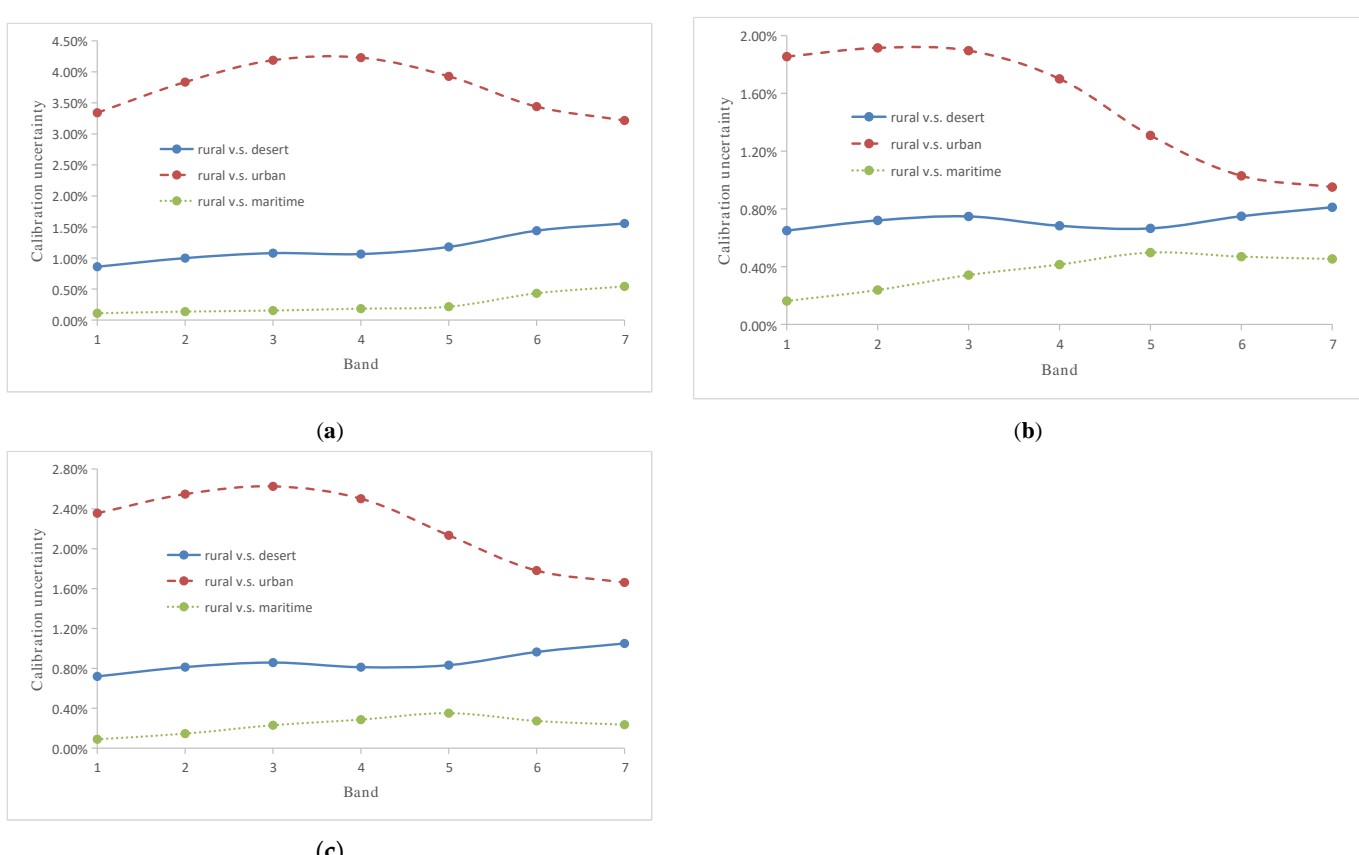

**Figure 13.** Calibration uncertainty caused by the assumption of different aerosol types for SDGSAT-1 MII calibration on 24 December 2021, utilizing the (**a**) reflectance-, (**b**) irradiance-, and (**c**) improved irradiance-based methods.

**Table 7.** Average relative differences in TOA radiance for the SDGSAT-1 MII.

| Method | Rural vs. Desert | Rural vs. Urban | Rural vs. Maritime |
|---|---|---|---|
| Reflectance-based method | 1.17% | 3.74% | 0.25% |
| Irradiance-based method | 0.72% | 1.52% | 0.37% |
| Improved irradiance-based method | 0.86% | 2.23% | 0.23% |

As shown in Figure 13 and Table 7, the urban aerosol type had the largest average relative difference from the rural aerosol type (3.74%), whereas the average relative differences for desert and maritime aerosol model assumptions were relatively minor, particularly for the maritime type (<0.54% in all seven bands). For the reflectance-based method, aerosol model assumption was the largest error source affecting the calibration accuracy, with the largest uncertainty of 3.74%. Both the irradiance- and improved irradiance-based methods adopt the DG ratio to reduce the uncertainty associated with aerosol model assumptions, and their uncertainties were significantly reduced to within 1.52% and 2.23%, respectively.

We further analysed the relative differences between the rural and the other three aerosol types for the reflectance-, irradiance-, and improved irradiance-based methods under different AOD conditions (550 nm AOD = 0.05, 0.1, 0.2, 0.3, 0.4, and 0.5) (Figure 14). The results confirmed that aerosol model assumption is the major factor affecting the calibration accuracy of the reflectance-based method under different AOD conditions. The urban aerosol type had the largest uncertainty when compared with the rural aerosol type under different AOD conditions, leading to the large uncertainty of the reflectance-based method. The irradiance-based method is the first choice for in-situ experiments for the desert and urban aerosol types. In contrast, the improved irradiance-based method is preferred for the rural and maritime aerosol types.

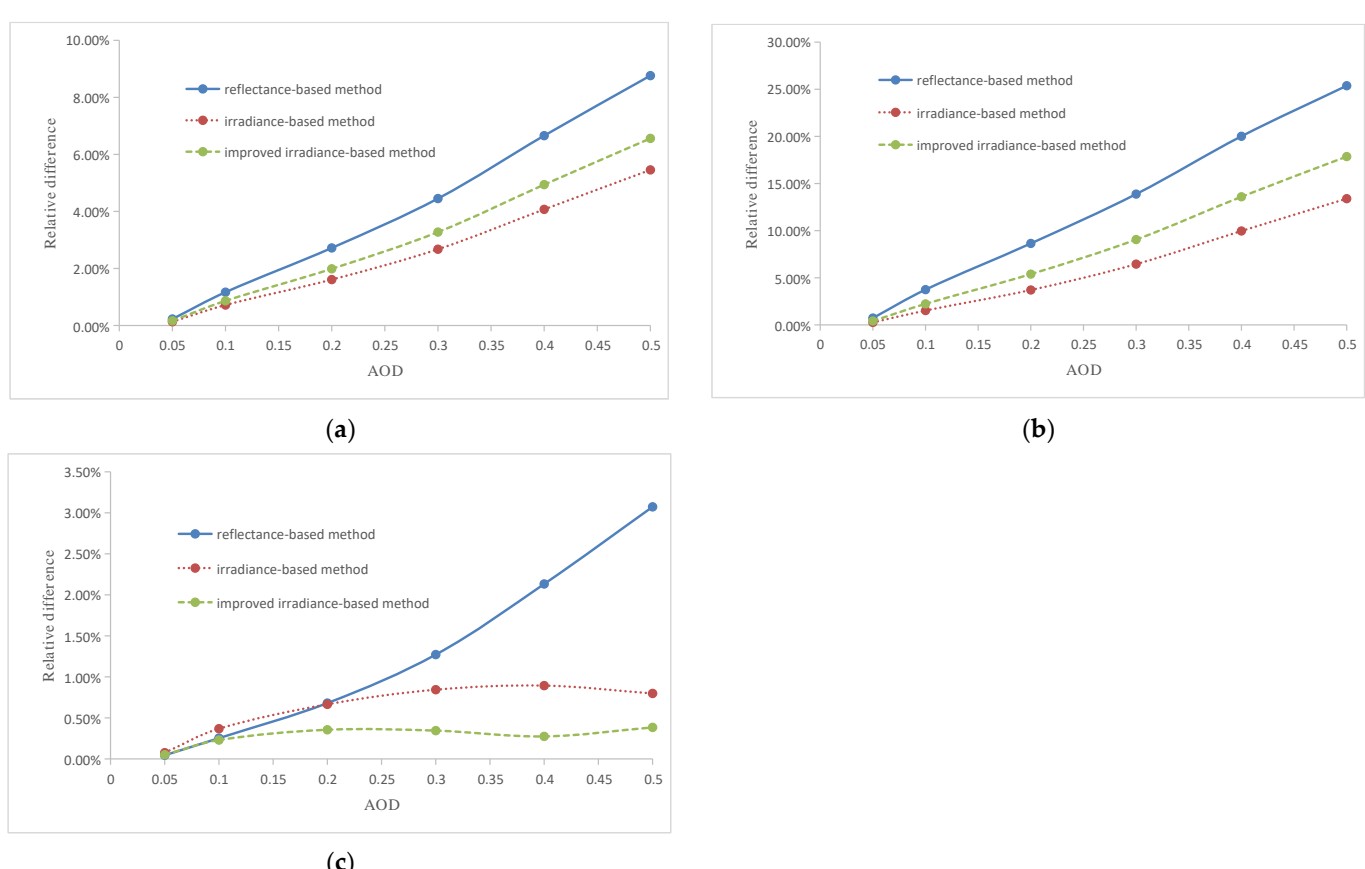

**Figure 14.** Relative differences between the (**a**) rural and desert aerosol types, (**b**) rural and urban aerosol types, and (**c**) rural and maritime aerosol types for the reflectance-, irradiance-, and improved irradiance-based methods under different AOD conditions (AOD = 0.05, 0.1, 0.2, 0.3, 0.4, and 0.5).

### 4.1.2. Uncertainty Analysis of Atmospheric Profile Measurements

Temperature, altitude, relative humidity, and air pressure data combined with other atmospheric components are utilized to obtain the TOA radiance in MODTRAN v.5.2.1. In this study, a radiosonde balloon was used to acquire the vertical atmospheric profile parameters as inputs for MODTRAN to simulate the TOA radiance. It should be noted

that there was a time difference of approximately 4 h between the radiosonde balloon release time (around 07:15 Beijing time) and the actual overpass time (around 11:45 Beijing time). To explore the uncertainty caused by atmospheric measurements, three MODTRAN-embedded atmospheric models (mid-latitude summer [MLS], mid-latitude winter, [MLW], and 1976 US standard atmosphere [US]) were used to simulate TOA radiances, which were compared with the results obtained by using the atmospheric profile measured by the radiosonde, as shown in Figure 15.

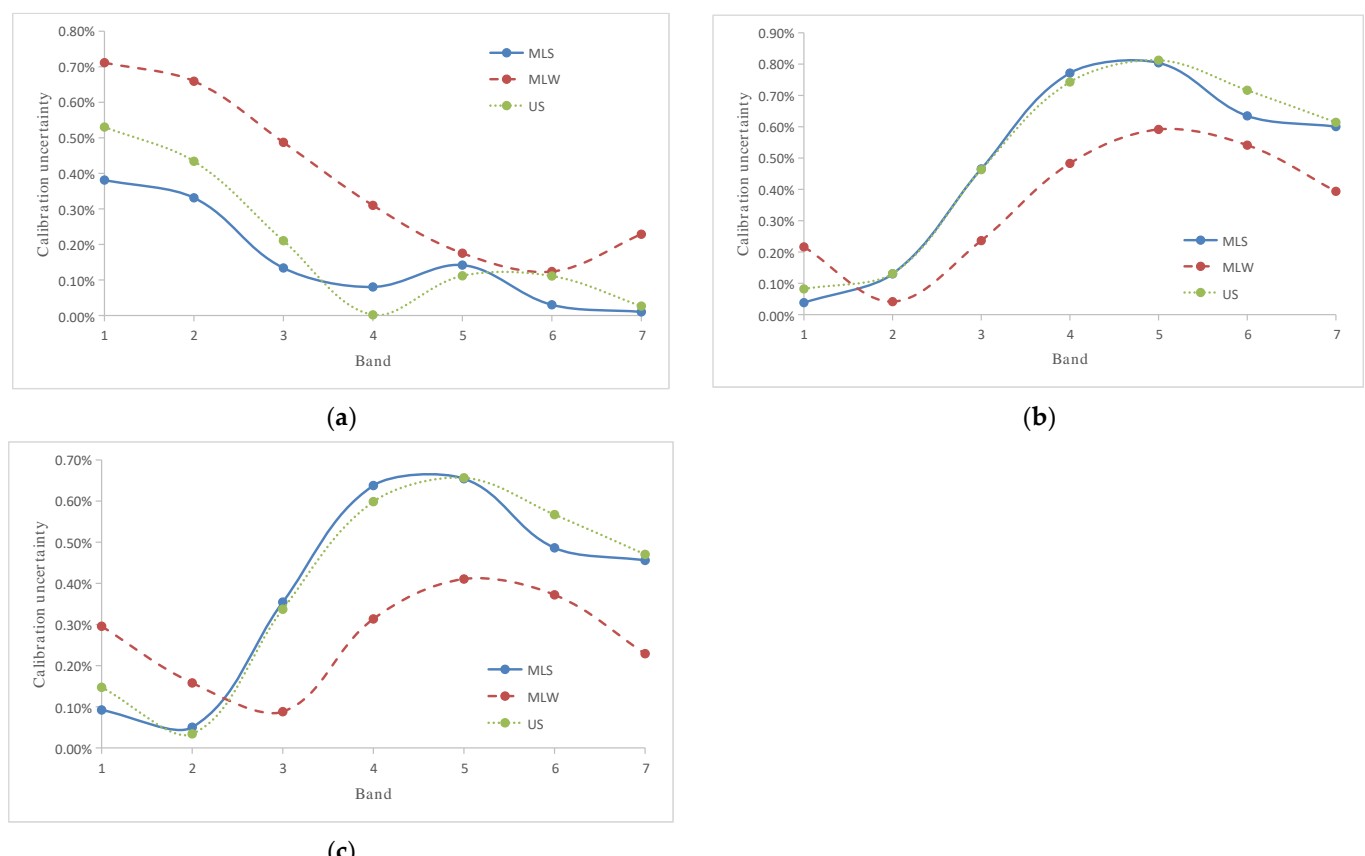

**Figure 15.** Calibration uncertainty caused by atmospheric profile measurements for SDGSAT-1 MII calibration on 24 December 2021, utilizing the (**a**) reflectance-, (**b**) irradiance-, and (**c**) improved irradiance-based methods.

In comparison to the reflectance-based method, the irradiance- and improved irradiance-based methods usually produced slightly higher uncertainties due to the atmospheric profile; however, the uncertainties in all bands of the two methods were below 0.81%. For the reflectance-based method, the predicted TOA using the MLS model appeared to be the closest to that obtained using the radiosonde measurements. In contrast, for the irradiance- and improved irradiance-based methods, the MLW model appeared to better reflect the radiosonde measurements and actual conditions. Therefore, we adopted the largest relative differences, derived from replacing the radiosonde measurements with the US and MLS models, as the vicarious radiometric calibration uncertainty caused by different atmospheric model assumptions.

### 4.1.3. Uncertainty Analysis of AOD Retrieval

The total uncertainty for the AOD, which is retrieved from CE318 data, is 0.01–0.02 [44]. In this experiment, the AOD at 550 nm was interpolated from CE318 measurements at 440 and 670 nm. As shown in Figure 5, the AOD at 550 nm at the Dunhuang radiometric calibration site varied from 0.096 to 0.155. To predict the calibration uncertainty owing to

variation in the AOD at 550 nm, the original AOD at 550 nm value of 0.1045 was replaced with values of 0.1245 and 0.0845 in radiative transfer simulation. The calibration uncertainty due to variation in the AOD at 550 nm was estimated by comparing the predicted TOA radiance at different levels with the base value set to 0.02. Figure 16 exhibits the calibration uncertainty owing to AOD measurements.

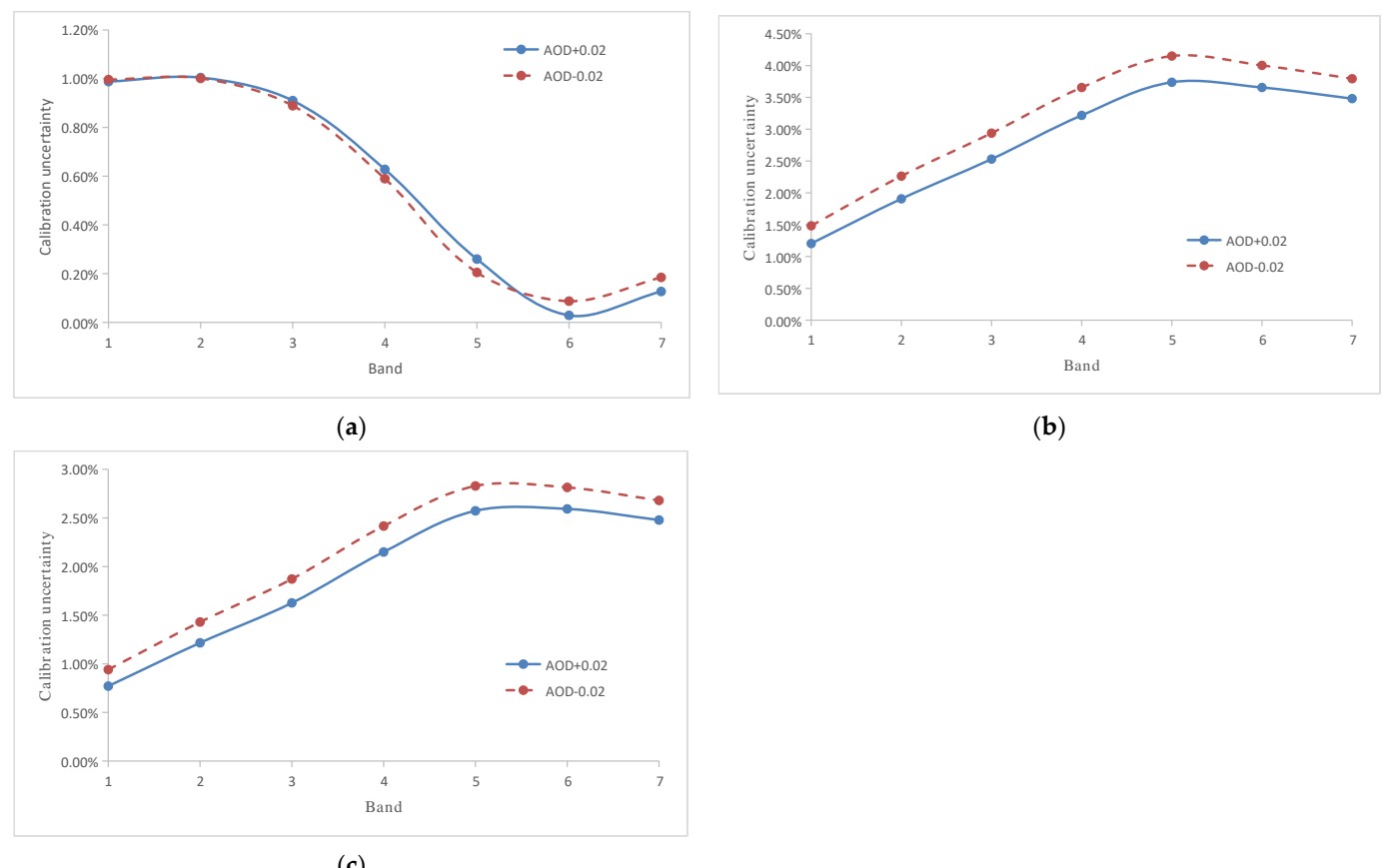

(**a**)

(**b**)

(**c**)

**Figure 16.** Calibration uncertainty caused by AOD retrieval for SDGSAT-1 MII calibration on 24 December 2021, utilizing the (**a**) reflectance-, (**b**) irradiance-, and (**c**) improved irradiance-based methods.

As shown in Figure 16, although the same difference was considered in the AOD measurements, the relative differences differed for the three methods. For the reflectance-based method, a variation of ±0.02 in the AOD added little uncertainty in the simulated TOA radiance, with a maximum value of 1.00%. In addition, when the AOD changed by ±0.02, the calibration uncertainty decreased with increasing wavelength. However, for the other two methods, a variation of ±0.02 in the AOD introduced significantly larger uncertainties in the simulated TOA radiance than for the reflectance-based method, with maximum values of 4.15% and 2.83%, respectively. Moreover, when the AOD changed by ±0.02, the calibration uncertainty became larger with increasing wavelength, which was the opposite of the uncertainty trend for the reflectance-based method.

A change in the AOD led to opposite trends for direct and diffuse transmittances, revealing an inverse relationship, and, as the same difference was considered in AOD retrieval, the relative difference was symmetrical (Figure 17). For the reflectance-based method (Equation (4)), both the direct transmittance and the diffuse transmittance ($T_{dir}(\theta_s)$, $T_{dif}(\theta_s)$, $T_{dir}(\theta_v)$, and $T_{dif}(\theta_v)$) were affected by AOD errors according to Equations (5) and (6). The direct transmittance increased or decreased with a change in the AOD, while the diffuse transmittance changed in the opposite direction, revealing a "when one falls, another rises" relation. This is why the calibration uncertainty caused by AOD errors

was relatively low when using the reflectance-based method, in which the variation in the direct transmittance partially compensates that of the diffuse transmittance in both downward and upward directions. As a comparison, the AOD variation mainly affected the direct transmittance in both downward and upward directions ($e^{-\tau/\mu_s}$ and $e^{-\tau/\mu_v}$, that is $T_{dir}(\theta_s)$ and $T_{dir}(\theta_v)$) in the irradiance-based method. With increasing wavelength, direct transmittances in both downward and upward directions also increased. The improved irradiance-based method is optimal for downward diffuse transmittance, as an additional compensating factor is included in the TOA radiance calculation. Consequently, the calibration uncertainty introduced by AOD errors is small for the reflectance-based method and large for the irradiance-based method compared to the improved irradiance-based method.

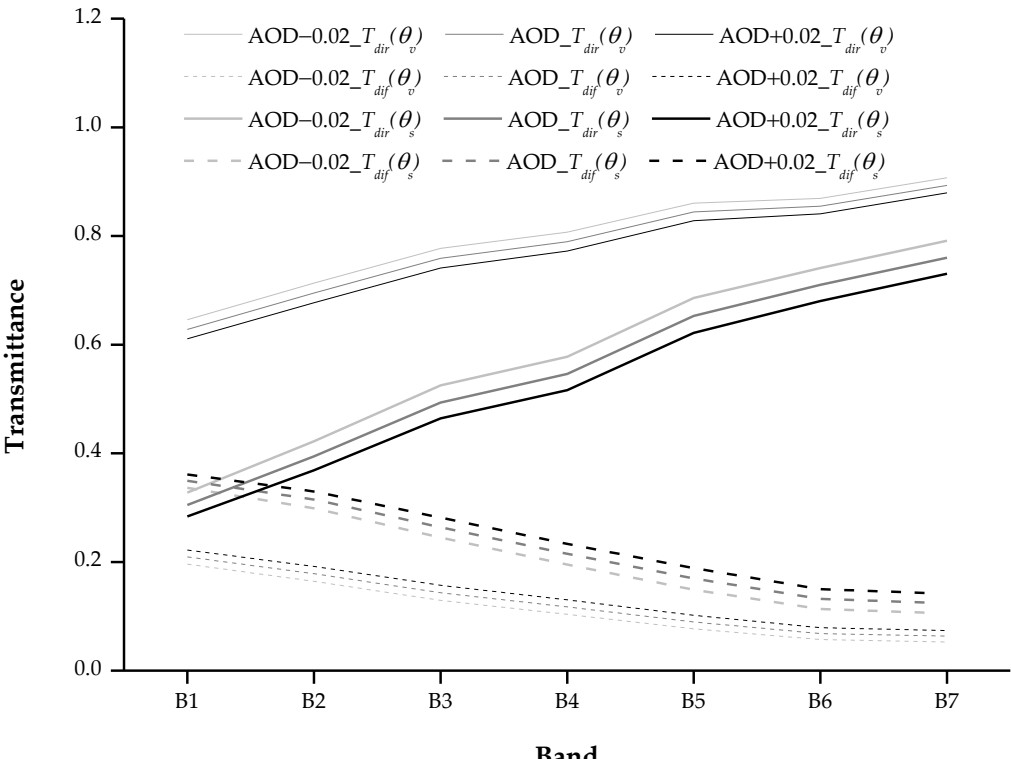

**Figure 17.** Changes in direct and diffuse transmittances for both downward and upward directions under different 550 nm AOD.

From Figure 17, we can infer that when the AOD is small, the irradiance- and improved irradiance-based methods are inferior to the reflectance-based method. To verify this, various AOD values (0.1, 0.2, 0.3, 0.4, and 0.5) were input into MODTRAN to obtain transmittance values (Figure 18). As expected, the direct transmittance decreased, while the diffuse transmittance increased with increasing AOD. The larger the AOD was, the smaller the direct transmittance and the larger the diffuse transmittance were. In addition, we calculated the corresponding transmittance errors when the AOD changed by ±0.02 under the different AOD conditions for the three methods, and the results (Table 8) confirmed our hypothesis. For the reflectance-based method, the transmittance error caused by an AOD change was significantly smaller than those for the irradiance- and the improved irradiance-based methods under different AOD conditions. Based on these findings, we concluded that when the AOD is low (≤0.1), the calibration accuracy of the reflectance-based method is higher than that of the irradiance- and improved irradiance-based methods.

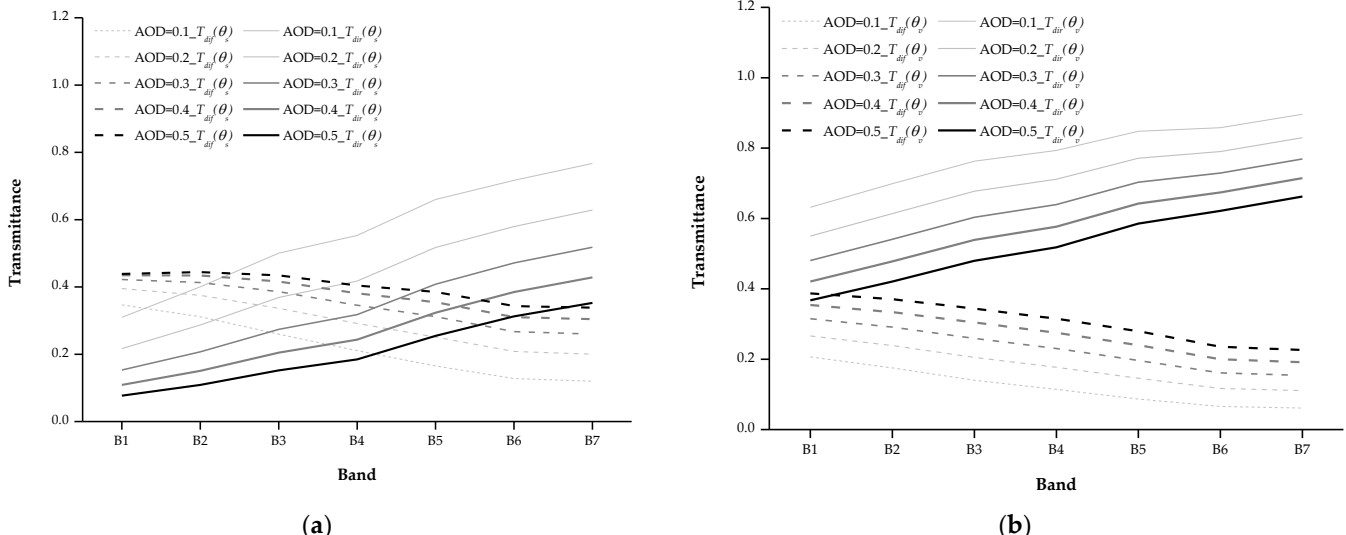

**Figure 18.** Changes in direct transmittance and transmittances under different 550 nm AOD conditions, including direct and diffuse transmittances in (**a**) downward and (**b**) upward directions.

**Table 8.** Transmittance errors when the AOD changes by $\pm 0.02$ under different AOD conditions for the reflectance-, irradiance-, and improved irradiance-based methods.

| AOD | | Reflectance-Based Method $(T(\theta_s) \times T(\theta_v))$ | Irradiance-Based Method $(T_{dir}(\theta_s) \times T_{dir}(\theta_v))$ | Improved Irradiance-Based Method $(T_{dir}(\theta_s) \times T(\theta_v))$ |
|---|---|---|---|---|
| AOD = 0.1 | AOD + 0.02 | 1.90% | 7.40% | 5.80% |
| | AOD − 0.02 | 1.97% | 8.05% | 6.20% |
| AOD = 0.2 | AOD + 0.02 | 1.77% | 7.33% | 5.75% |
| | AOD − 0.02 | 1.83% | 7.95% | 6.12% |
| AOD = 0.3 | AOD + 0.02 | 1.62% | 6.93% | 5.44% |
| | AOD − 0.02 | 1.65% | 7.39% | 5.72% |
| AOD = 0.4 | AOD + 0.02 | 1.59% | 7.15% | 5.62% |
| | AOD − 0.02 | 1.62% | 7.68% | 5.94% |
| AOD = 0.5 | AOD + 0.02 | 1.55% | 7.29% | 5.74% |
| | AOD − 0.02 | 1.58% | 7.86% | 6.08% |

4.1.4. Uncertainty Analysis of Water Vapor Measurement

Like the AOD at 550 nm, the CWV is retrieved from CE318 data, and the error of CWV retrieval is within 10% [33]. Therefore, an uncertainty of $\pm 10\%$ was replaced with the CWV measured by CE318 in MODTRAN 5.2.1. As shown in Figure 19, the effect of a change in CWV on calibration uncertainty was nearly negligible in all bands, with a maximum value of 0.32%. Given that the Dunhuang calibration site is located in an arid area with dry atmospheric conditions, the water vapor measurement errors have little impact on the calibration uncertainty.

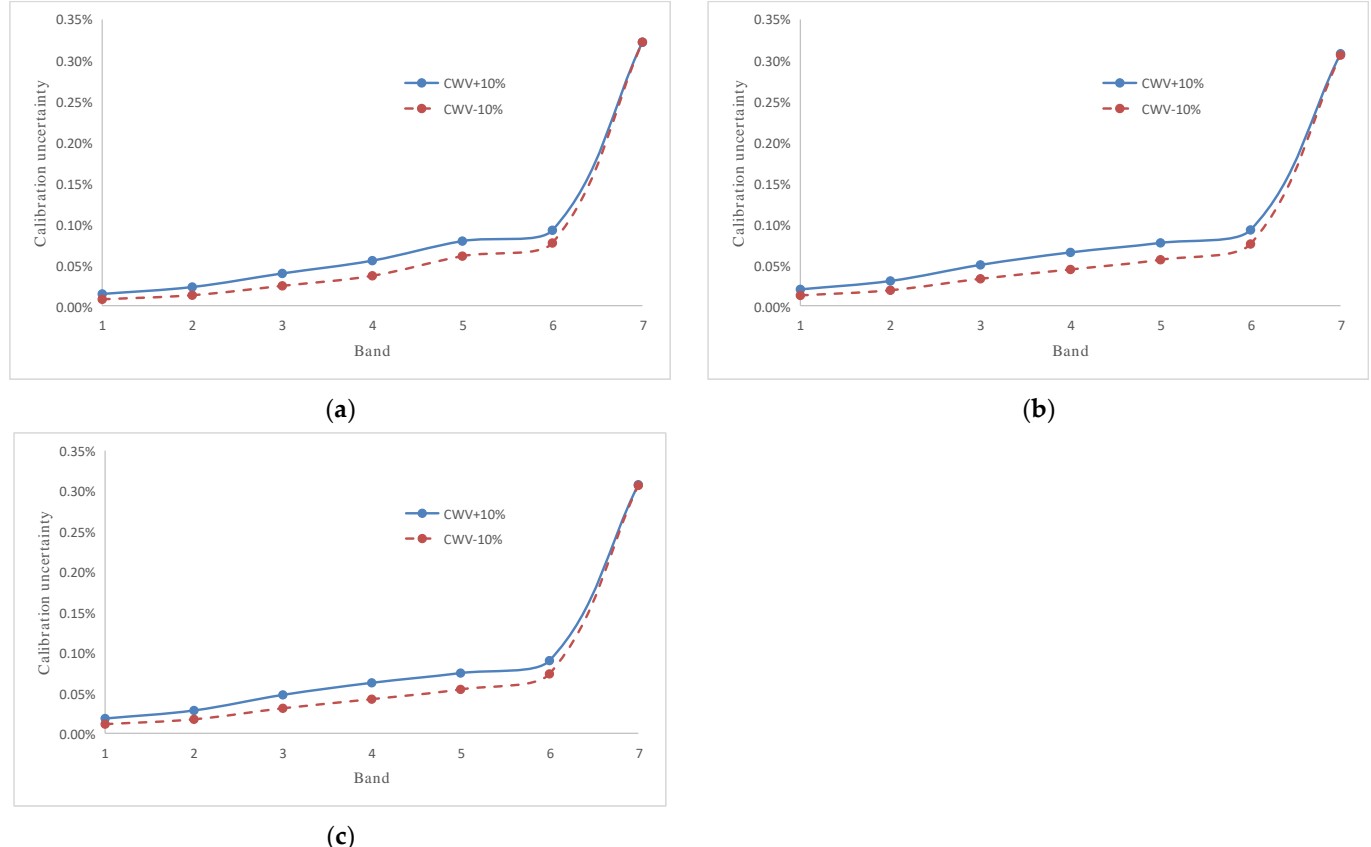

**Figure 19.** Calibration uncertainty caused by CWV retrieval for SDGSAT-1 MII calibration on 24 December 2021, utilizing the (**a**) reflectance-, (**b**) irradiance-, and (**c**) improved irradiance-based methods.

### 4.1.5. Uncertainty Analysis of Simultaneous Ground Reflectance Measurements

The Dunhuang calibration site is one of China's main radiometric calibration sites for satellites with visible to near-infrared band sensors [2]. The stable ground reflectance of the site guarantees high radiometric calibration accuracy. Both the reflectance- and irradiance-based methods rely on simultaneous ground reflectance measurements. The accuracy of the simultaneous ground reflectance measurement, as MODTRAN simulation input, directly affects the radiative transfer accuracy. Multiple calibration studies have indicated that the error of ground reflectance measurement is approximately 2% [33,43,45,46]. The ground reflectance at the Dunhuang calibration site fluctuated by ±1.5% in the last 15 years, showing good stability [47]. In addition, the measurement procedure and instrument specifications meet the requirements of in-situ calibration experiments; therefore, the calibration uncertainty caused by ground reflectance errors was set to 1.5% in this study.

### 4.1.6. Uncertainty Analysis of Viewing Geometry

The solar zenith angle, solar azimuth angle, view zenith angle, and view azimuth angle are all parts of the satellite viewing geometry. They determine the amount of energy reflected from the ground and the energy received by the sensor entrance pupil. The solar angles can be calculated according to the satellite transit time and the longitude and latitude of the calibration site with relatively high accuracy. The viewing angles can be obtained from information in the image itself. To assess the uncertainty brought by the viewing geometry, we assumed that the calculation errors of the solar angles were 0.1° because of high calculation precision, whereas those of the viewing angles were set to 1° [42]. By sequentially changing the solar and viewing angles, the TOA radiance under the new viewing geometry was obtained and compared with the actual observation results. The relative differences between the simulated results under different angles and the

actual observation results, calculated using Equation (14), were regarded as the calibration uncertainty caused by a variation in the viewing geometry (Figure 20).

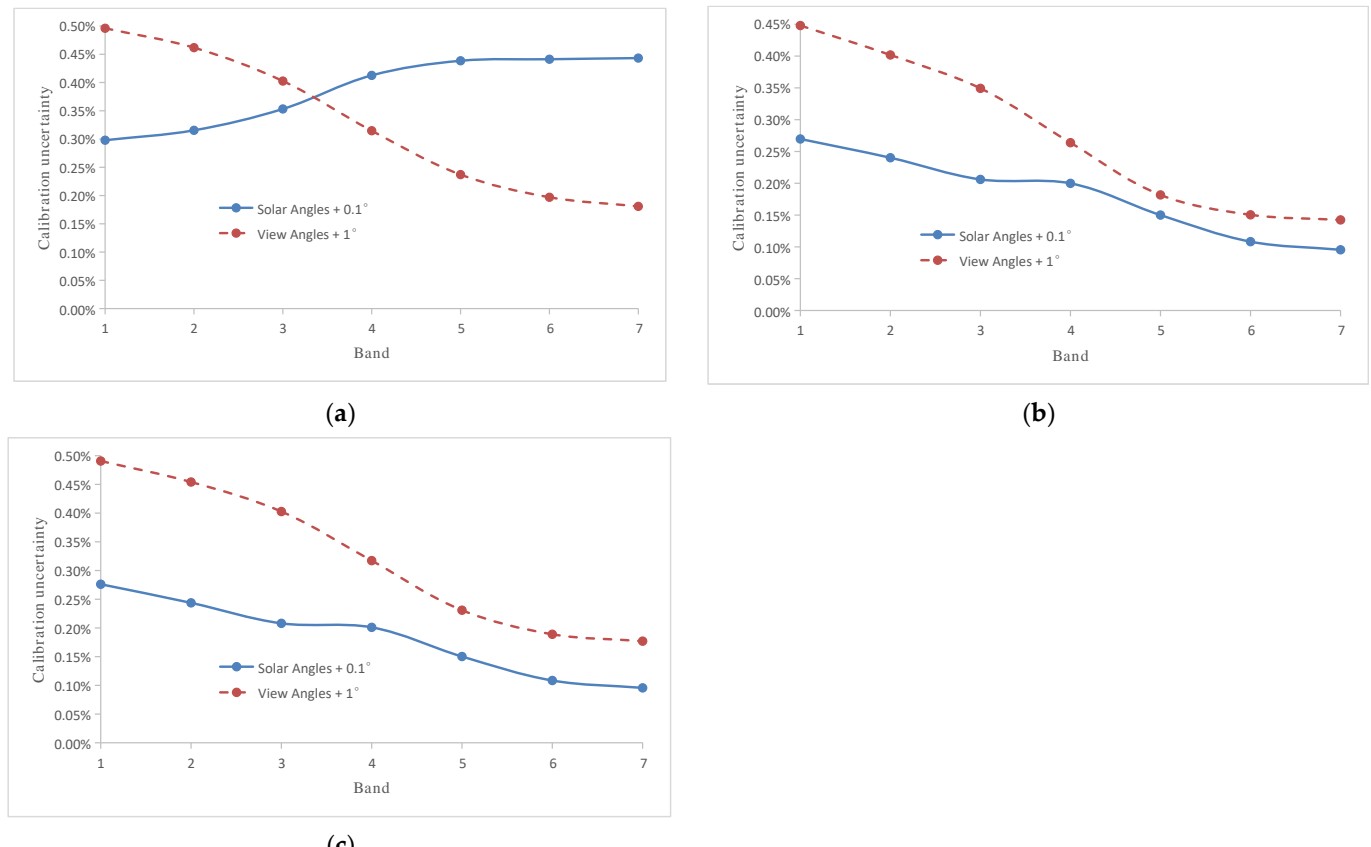

**Figure 20.** Calibration uncertainty caused by viewing geometry errors for SDGSAT-1 MII calibration on 24 December 2021, utilizing the (**a**) reflectance-, (**b**) irradiance-, and (**c**) improved irradiance-based methods.

Changes in the solar and viewing angles had little impact on the TOA radiance. When the solar angles increased by 0.1°, the variation in the TOA radiance was below 0.44%. When the viewing angles increased by 1°, the variation in the TOA radiance was below 0.50%. Therefore, the calibration uncertainty caused by viewing geometry errors was set to 0.50%.

### 4.1.7. Total Calibration Uncertainty Estimation

Previous studies have demonstrated that the calibration uncertainties that come from other sources are relatively stable [29,31,33,48,49]. The calibration uncertainties caused by the ozone measurement [49,50], BRDF error [36], image misregistration errors, radiative transfer code accuracy [51,52], and DG ratio measurement [48] were estimated to be 0.6%, 2.0%, 0.2%, 1.0%, and 2.0%, respectively. The total vicarious radiometric calibration uncertainty caused by all of the above factors was calculated according to Equation (12). The total calibration uncertainties for the three methods are presented in Table 9.

According to previous studies, the accuracy of the irradiance- and improved irradiance-based methods is generally better than that of the reflectance-based method in the case of large AOD. However, the data in Table 9 makes it abundantly evident that the total uncertainties of both the irradiance- and improved irradiance-based methods were slightly higher than that of the reflectance-based method. This is because the DG ratio is an indispensable parameter in the former methods. In addition to the same parameters as the reflectance-based method, the irradiance-based method also uses DG ratios in both the

solar and viewing directions. However, it is difficult to directly measure the DG ratio in the viewing direction, which has to be extrapolated by fitting the values under different solar zenith angles. During the calibration experiment on 14 December 2021, because of the small variation range of the solar zenith angle within the effective observation time, it was difficult to obtain a relatively accurate DG ratio in the viewing direction after linear extrapolation, which is one of the main reasons why the total uncertainty of the irradiance-based method was higher than that of the reflectance-based method in this study. In the improved irradiance-based method, only the DG ratio measured in the solar direction is included in the calculation to avoid the error introduced by linear extrapolation. Therefore, this method is more accurate than the irradiance-based method. Overall, the irradiance- and improved irradiance-based methods strongly rely on the precision of the direct transmittance measurements and the DG ratios; consequently, improving their measurement accuracy is key to high-precision calibration.

**Table 9.** Absolute calibration uncertainties of reflectance-, irradiance-, and improved irradiance-based methods.

| Source of Uncertainty | Reflectance-Based Method (%) | Irradiance-Based Method (%) | Improved Irradiance-Based Method (%) |
|---|---|---|---|
| Assumption of aerosol type | 0.11–4.23 | 0.16–1.91 | 0.09–2.63 |
| Assumption of atmospheric model | 0.01–0.71 | 0.04–0.81 | 0.09–0.66 |
| 550 nm AOD retrieval | 0.03–1.00 | 1.20–4.15 | 0.77–2.83 |
| Water vapor retrieval | 0.01–0.32 | 0.01–0.31 | 0.01–0.31 |
| Ozone measurement | 0.6 | 0.6 | 0.6 |
| Ground reflectance measurement | 1.5 | 1.5 | 1.5 |
| BRDF error | 2.0 | 2.0 | 2.0 |
| Viewing geometries | 0.18–0.50 | 0.10–0.45 | 0.10–0.49 |
| Image misregistration errors | 0.2 | 0.2 | 0.2 |
| Radiative transfer code accuracy | 1.0 | 1.0 | 1.0 |
| DG ratio measurement | / | 2.0 | 2.0 |
| Total uncertainty (Root sum of squares) | 2.77–5.23 | 3.62–5.79 | 3.50–5.23 |

The above results were obtained under a low AOD, and we concluded that the largest calibration uncertainty in the reflectance-based method stemmed from the aerosol model assumption, whereas the retrieval uncertainty of the 550 nm AOD was the primary calibration uncertainty for the other two methods. We analysed and discussed the reasons for this conclusion in Sections 4.1.1 and 4.1.3, respectively. The actual aerosol type in Dunhuang is similar to the rural and desert types, and the uncertainty of the aerosol model assumption was expected to not lead to such a significant relative discrepancy between the urban and rural aerosol types. Therefore, we adopted the relative difference between the rural and desert aerosol types as the uncertainty of the aerosol model assumption. The total uncertainties of the three methods under different AOD conditions are provided in Table 10. When the AOD was 0.1, the reflectance-based method performed better than the other two methods, with a total uncertainty of 3.10%. When the AOD was 0.2, the irradiance-based method performed slightly better than the other two methods, with a total uncertainty of 3.84%, and the accuracy of the reflectance-based method was basically the same as that of the improved irradiance-based method. The irradiance-based method performed best when the AOD was 0.3, 0.4, or 0.5.

**Table 10.** The total uncertainties of reflectance-, irradiance-, and improved irradiance-based methods under different AOD conditions.

| Total Uncertainty | Reflectance-Based Method (%) | Irradiance-Based Method (%) | Improved Irradiance-Based Method (%) |
|---|---|---|---|
| AOD = 0.1 | 3.10 | 3.55 | 3.58 |
| AOD = 0.2 | 3.96 | 3.84 | 4.00 |
| AOD = 0.3 | 5.28 | 4.37 | 4.76 |
| AOD = 0.4 | 7.24 | 5.35 | 6.03 |
| AOD = 0.5 | 9.21 | 6.47 | 7.42 |

The three methods each have their advantages and applicable conditions. Our research results suggest that when the 550 nm AOD is low ($\leq$0.1), the calibration accuracy of the reflectance-based method may be higher than that of the other two methods, but when the 550 nm AOD becomes higher (>0.1), the irradiance-based method is the first choice. Therefore, it is essential to use multiple separate methods to calibrate a sensor in a vicarious calibration experiment, to mutually verify the results and detect possible errors caused by a single method. Furthermore, if conditions permit, calibration experiments should be repeated as often as possible to reduce the uncertainty of calibration accuracy.

### 4.2. Validation

Validation, i.e., testing and affirming the calibration results, is an important step in calibration experiments. Only after radiometric calibration and validation research can data products obtained from remote sensing inversion be credible and used in quantitative applications. In this section, we first compare the reflectance retrieved using the FLAASH atmospheric correction software with the in-situ measured reflectance over the Dunhuang calibration site. Then we utilize the well-calibrated Landsat-8 OLI and Sentinel-2A MSI as reference sensors to cross-validate the SDGSAT-1 MII results over different surface types. Finally, we calculate the relative difference in the retrieved ground reflectance between MII and the other two sensors.

### 4.2.1. Comparison with Measured Reflectance over the Dunhuang Calibration Site

We used the reflectance inversion method [53] to verify the rationality of the radiometric calibration methods in this paper. The calibration coefficients obtained in Section 3 were used in conjunction with FLAASH to obtain the inverse reflectance, which was then compared with the measured reflectance over the Dunhuang calibration site. A comparison of the reflectance inversion results is shown in Table 11. The inversed reflectance was largely consistent with the measured reflectance, and the absolute difference between them was within 0.001, demonstrating the rationality of the calibration methods.

**Table 11.** Comparison of reflectance inversion results.

| Band | Inversion | Measurement | Absolute Difference |
|---|---|---|---|
| B1 | 0.136456 | 0.136229 | 0.00023 |
| B2 | 0.155277 | 0.154839 | 0.00044 |
| B3 | 0.175996 | 0.177304 | 0.00131 |
| B4 | 0.206255 | 0.207094 | 0.00084 |
| B5 | 0.233118 | 0.233125 | 0.00001 |
| B6 | 0.236585 | 0.236509 | 0.00008 |
| B7 | 0.233967 | 0.233836 | 0.00013 |

#### 4.2.2. Ground Reflectance Validation Cross-Compared with Landsat-8 OLI and Sentinel-2A MSI over Different Surface Types

Because of the lack of spectral measurements of typical features, we utilized remote sensors with high radiometric calibration accuracy as references to cross-validate the sensor to be calibrated. We selected the Landsat-8 OLI and Sentinel-2A MSI as reference sensors to cross-validate the SDGSAT-1 MII. The details of the near-coincident Landsat-8 OLI and Sentinel-2A MSI images are provided in Table 12. The MII, OLI, and MSI overpass times over the Dunhuang calibration site differ. Therefore, we had to consider the influence of atmospheric conditions. The retrieved ground reflectance was obtained using the atmospheric parameters measured at the Dunhuang calibration site and the FLAASH atmospheric model according to the different radiometric calibration coefficients.

**Table 12.** Near-coincident Landsat-8 OLI and Sentinel-2A MSI data used in this study.

| Sensor | Data Acquisition Time (UTC + 8) | Center Coordinate | Solar Zenith | Solar Azimuth | 550 nm AOD | CWV (g/cm$^2$) |
|---|---|---|---|---|---|---|
| SDGSAT-1 MII | 14 December 2021, 11:45:17 | 40.0913°N, 94.3938°E | 68.5021° | 152.2956° | 0.1045 | 0.3114 |
| Landsat-8 OLI | 14 December 2021, 12:20:23 | 40.3329°N, 95.0782°E | 65.8780° | 161.3918° | 0.1199 | 0.2899 |
| Sentinel-2A MSI | 17 December 2021, 12:32:11 | 40.1419°N, 94.8185°E | 65.0575° | 164.5149° | 0.1229 | 0.2880 |

Figure 21 shows the relative spectral response curves of comparable bands for SDGSAT-1 MII, Landsat-8 OLI, and Sentinel-2A MSI. Table 13 shows the visible and near-infrared band information for the three sensors. To directly compare the spectral ground reflectance retrieved by FLAASH between SDGSAT-1 MII, Landsat-8 OLI, and Sentinel-2A MSI, the spectral band differences must be accounted for according to Equations (16) and (17).

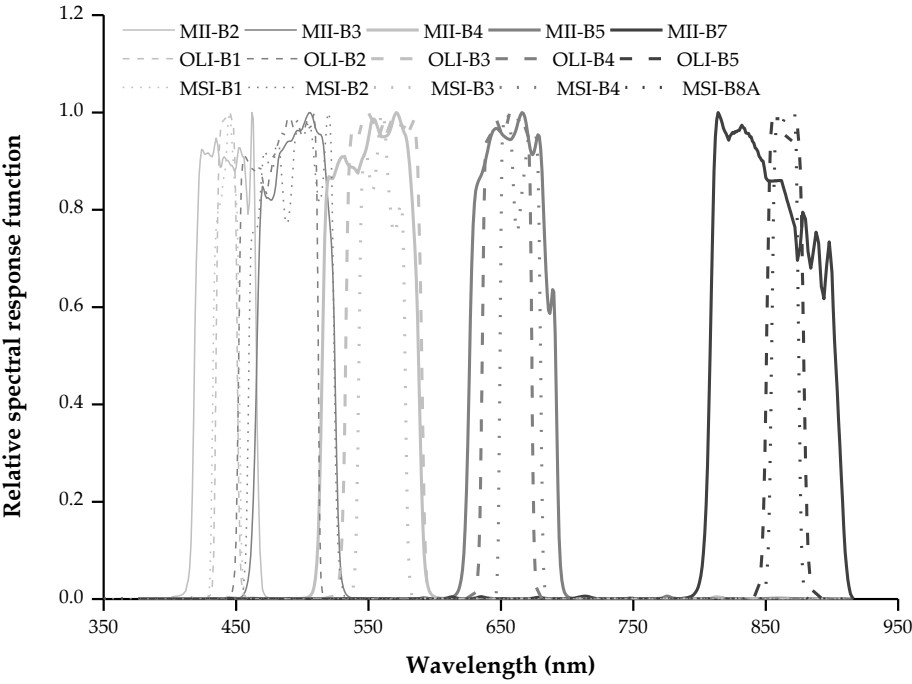

**Figure 21.** Relative spectral responses of corresponding bands for SDGSAT-1 MII, Landsat-8 OLI, and Sentinel-2A MSI.

**Table 13.** Spectral ranges of SDGSAT-1 MII, Landsat-8 OLI, and Sentinel-2A MSI.

| SDGSAT-1 MII | | | Landsat-8 OLI | | | Sentinel-2A MSI | | |
|---|---|---|---|---|---|---|---|---|
| Band | Center Wavelength (nm) | Spectral Range (nm) | Band | Center Wavelength (nm) | Spectral Range (nm) | Band | Center Wavelength (nm) | Spectral Range (nm) |
| B2 | 462 | 410–467 | B1 | 445 | 435–451 | B1 | 443 | 431–454 |
| B3 | 506 | 457–529 | B2 | 509 | 452–512 | B2 | 520 | 458–527 |
| B4 | 571 | 510–597 | B3 | 550 | 533–590 | B3 | 560 | 504–602 |
| B5 | 666 | 618–696 | B4 | 656 | 636–673 | B4 | 654 | 649–680 |
| B7 | 814 | 798–911 | B5 | 859 | 851–879 | B8A | 871 | 855–875 |

Different surface types, including water, bare land, desert, and dry farmland, around Dunhuang City in Gansu Province, China, were selected to evaluate the vicarious calibration coefficients according to the reflectance-, irradiance-, and improved irradiance-based methods. Figure 22 shows the water, bare land, desert, and dry farmland sites selected for measurements on 14 December 2021. The ground reflectance in the regions of interest was calculated based on the different calibration coefficients, measured atmospheric parameters, and FLAASH. Using this approach, the vicarious radiometric calibration coefficients of SDGSAT-1 MII were validated for all selected surface types, as shown in the scatter plots and curve fits in Figure 23.

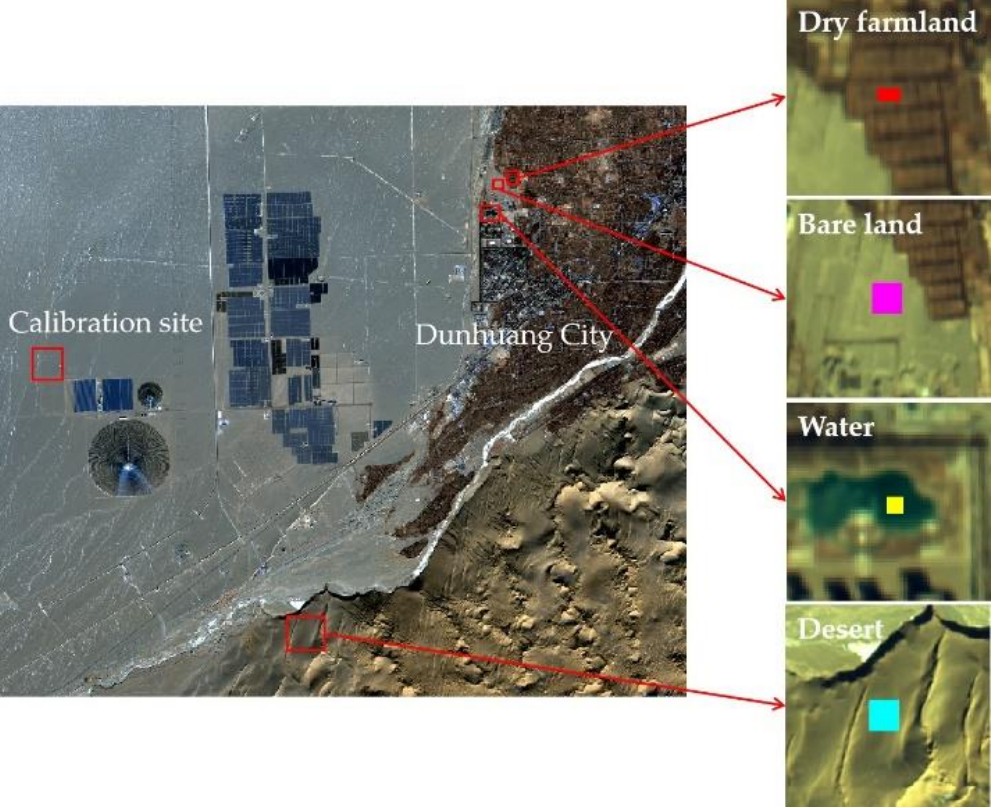

**Figure 22.** Different surface types selected for validation.

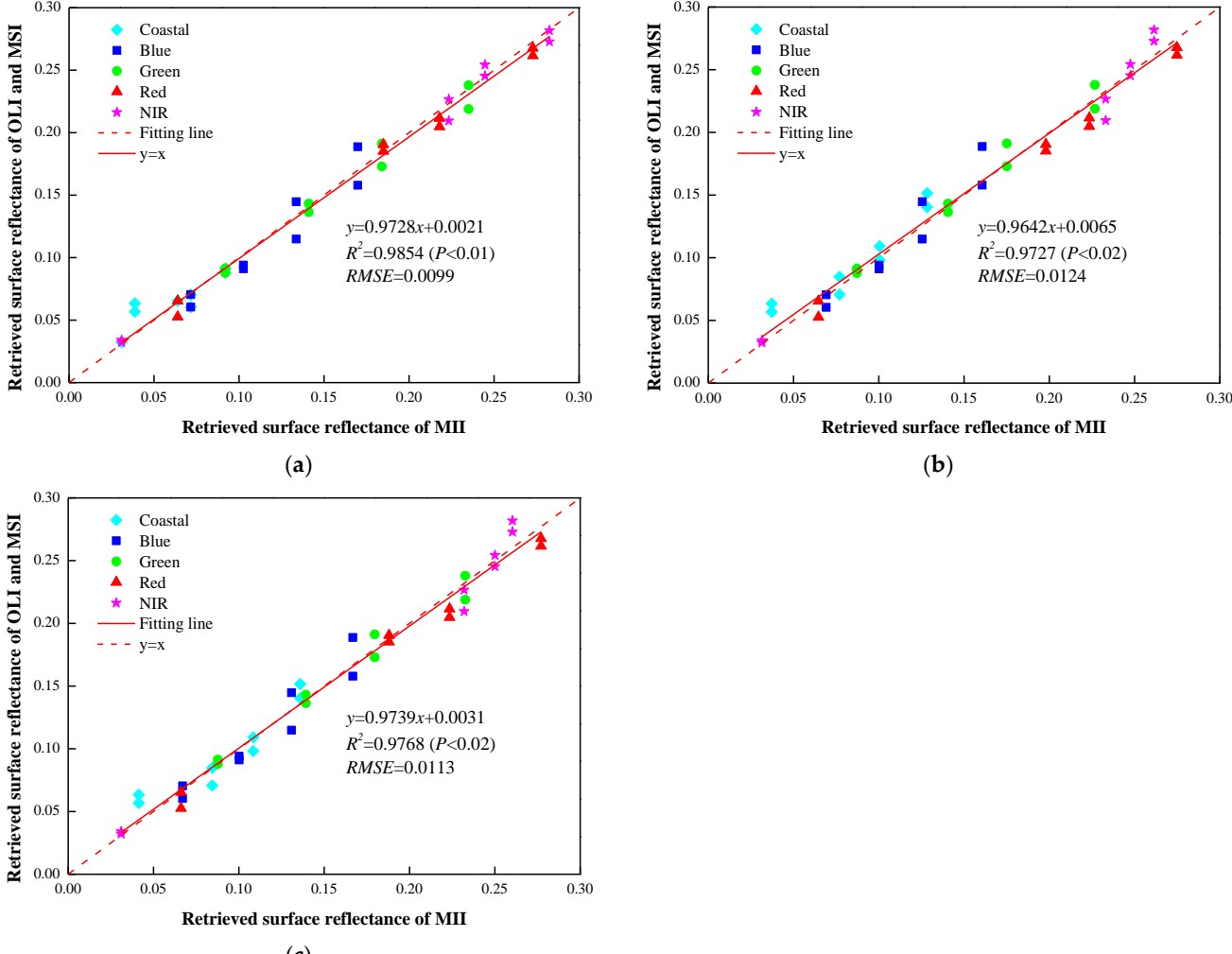

**Figure 23.** Comparison of retrieved ground reflectances obtained by the indicated sensors according to the (**a**) reflectance-, (**b**) irradiance-, and (**c**) improved irradiance-based methods.

Figure 23 shows the cross-comparison results of the reflectances for the selected surface types obtained using the different satellite sensors and the reflectance-, irradiance-, and improved irradiance-based methods. y = x is the reference line; the closer the data points are to y = x, the closer the retrieved ground reflectance of MII is to that of OLI and MSI and, thus, the higher accuracy of the calibration method. All data points were linearly fitted, and the linear fitting equation expression and goodness-of-fit ($R^2$) were obtained. The larger the $R^2$ value, the higher the degree of fit. In addition, the root mean square error (RMSE) of each calibration method was calculated. The smaller the RMSE, the higher the accuracy of the calibration method.

Overall, the retrieved ground reflectance of the MII was in good agreement with that of the OLI and MSI sensors for all three methods ($R^2 > 97\%$, $p < 0.02$, RMSE < 0.013). The results revealed that the calibrated SDGSAT-1 MII image was highly consistent with the Landsat-8 OLI and Sentinel-2A MSI images. Interestingly, the calibration accuracy of the three methods was also reflected by the $R^2$ values (reflectance-based, 0.9854, $p < 0.01$; improved irradiance-based, 0.9768, $p < 0.02$; and irradiance-based, 0.9727, $p < 0.02$) and RMSE values (0.0124, 0.0113, and 0.0099, respectively). The cross-comparison results agreed with the results exhibited in Section 4.1, with the reflectance-based method having a slightly higher accuracy than the other two methods.

To quantitatively verify the calibration accuracy of the different methods, the absolute and relative differences between MII and OLI/MSI were calculated. Table 14 lists the

results. The absolute (relative) differences in most bands were below 0.009 (5%). The average absolute (relative) differences between MII and OLI and between MII and MSI were all within 0.01 (10%). This demonstrated that the calibration accuracy of the MII coincides with that of the OLI and MSI sensors, which is consistent with the above results.

**Table 14.** Average absolute (relative) differences in retrieved reflectances over different surface types between MII and OLI/MSI utilizing the reflectance-, irradiance-, and improved irradiance-based methods.

| Method | Band | Absolute (Relative) Difference between MII and OLI | Absolute (Relative) Difference between MII and MSI |
|---|---|---|---|
| Reflectance-based method | Coastal/aerosol | 0.0111 (14.04%) | 0.0086 (12.84%) |
| | Blue | 0.0133 (13.76%) | 0.0099 (7.12%) |
| | Green | 0.0081 (4.46%) | 0.0041 (2.78%) |
| | Red | 0.0033 (1.83%) | 0.0103 (8.82%) |
| | Near infrared | 0.0016 (1.57%) | 0.0091 (5.81%) |
| | Average difference | 0.0075 (7.13%) | 0.0084 (7.47%) |
| Irradiance-based method | Coastal/aerosol | 0.0165 (18.45%) | 0.0101 (13.56%) |
| | Blue | 0.0078 (8.86%) | 0.0137 (9.10%) |
| | Green | 0.0046 (3.16%) | 0.0076 (3.92%) |
| | Red | 0.0082 (4.09%) | 0.0129 (10.28%) |
| | Near infrared | 0.0074 (3.31%) | 0.0111(6.38%) |
| | Average difference | 0.0089 (7.57%) | 0.0111 (8.65%) |
| Improved irradiance-based method | Coastal/aerosol | 0.0097 (15.54%) | 0.0110 (15.04%) |
| | Blue | 0.0101 (10.12%) | 0.0113 (8.13%) |
| | Green | 0.0068 (4.09%) | 0.0052 (2.78%) |
| | Red | 0.0062 (2.95%) | 0.0126 (10.54%) |
| | Near infrared | 0.0082 (3.94%) | 0.0106 (6.48%) |
| | Average difference | 0.0082 (7.33%) | 0.0101 (8.59%) |

The average absolute (relative) differences between MII and OLI were in the order reflectance-based method (0.0075 (7.13%)) < improved irradiance-based method (0.0082 (7.33%)) < irradiance-based method (0.0089 (7.57%)); those between MII and MSI were in the order reflectance-based method (0.0084 (7.47%)) < improved irradiance-based method (0.0101 (8.59%)) < irradiance-based method (0.0111 (8.65%)). The absolute (relative) differences in the retrieved ground reflectance between MII and OLI/MSI provided strong evidence that the calibration accuracy of the reflectance-based method is basically the same as that of irradiance- and improved irradiance-based methods, and it is even better when the AOD is low ($\leq$0.1).

Notably, the relative differences were the largest in the coastal/aerosol and blue bands for all three methods (all > 10%). This is partly attributed to the low reflectance of water and the low response of the blue channel; therefore, a small difference will be amplified. Moreover, the relative difference in the red band was approximately 10% due to the low reflectance of water, which introduces a large error, resulting in a large difference over all surface types in the red band.

Overall, our research revealed that the calibration accuracy of the SDGSAT-1 MII is highly consistent with that of the Landsat-8 OLI and Sentinel-2A MSI, and we demonstrated that the calibration coefficients for the SDGSAT-1 MII are reliable and highly accurate. The different calibration methods were found to have different calibration accuracy; when the AOD is low ($\leq$0.1), the calibration accuracy of the reflectance-based method is similar to or even higher than that of the irradiance- and improved irradiance-based methods.

## 5. Conclusions

We comprehensively described the first in-situ vicarious radiometric calibration experiment of the SDGSAT-1 MII at the Dunhuang calibration site on 14 December 2021. In-situ

measurements, including ground reflectance, atmospheric parameters, and radiosonde data, were acquired during the satellite overpass date. Reflectance-, irradiance-, and improved irradiance-based calibration methods were utilized to predict the TOA spectral radiances using MODTRAN v.5.2.1 software. The vicarious radiometric calibration coefficients for the SDGSAT-1 MII were directly determined by dividing the TOA spectral radiances by the averaged DN over the $500 \times 500$ m calibration site. The calibration uncertainties were analysed in detail in this paper. The key findings are summarized as follows:

(1) The radiometric calibration coefficients obtained by the three vicarious calibration methods were reliable, with average relative differences of 2.20% (between the reflectance- and irradiance-based methods) and 1.43% (between the reflectance- and improved irradiance-based methods). The total calibration uncertainties of the reflectance-, irradiance-, and improved irradiance-based methods were 2.77–5.23%, 3.62–5.79%, and 3.50–5.23%, respectively. The largest calibration uncertainty in the reflectance-based method stemmed from the aerosol model assumption, while the retrieval uncertainty of the AOD at 550 nm was the primary calibration uncertainty in the other two methods.

(2) To verify the calibration accuracy of SDGSAT-1 MII, we utilized well-calibrated sensors, Landsat-8 OLI and Sentinel-2A MSI, as reference sensors for cross-comparison with the SDGSAT-1 MII over different surface types. MII retrievals differed less than 0.0075 (7.13%) from OLI retrievals and less than 0.0084 (7.47%) from MSI retrievals when applying calibration coefficients from the reflectance-based method; less than 0.0089 (7.57%) from OLI retrievals and less than 0.0111 (8.65%) from MSI retrievals for the irradiance-based method; and less than 0.0082 (7.33%) from OLI retrievals and less than 0.0101 (8.59%) from MSI retrievals for the improved irradiance-based method. The cross-comparison results showed that the SDGSAT-1 MII calibration accuracy was consistent with those of Landsat-8 OLI and Sentinel-2A MSI, and they showed that the calibration coefficients for the SDGSAT-1 MII are reliable and highly accurate.

(3) Unexpected calibration errors caused by measurement uncertainties in in-situ calibration experiments are hard to detect. Therefore, using different calibration methods and cross-comparing their results is important. In the case of a low AOD ($\leq$0.1), the irradiance- and improved irradiance-based methods showed no obvious advantage over the reflectance-based method, which is therefore recommended under such conditions due to its superiorities of easy operations, low cost, and high accuracy. If DG ratios are not measured, using only the reflectance-based method for calibration under ideal weather conditions is reliable.

(4) The performances of the three calibration methods varied for different aerosol types under different AOD conditions. For calibration sites with ground and atmospheric conditions similar to the Dunhuang calibration site, we suggest using the rural and desert aerosol types. When the AOD was 0.1, the reflectance-based method was the best choice, with a total uncertainty of 3.10%, whereas when the AOD was 0.2, 0.3, 0.4, or 0.5, the irradiance-based method had a higher accuracy, with total uncertainties of 3.84%, 4.37%, 5.35%, and 6.47%, than the other two methods. The improved irradiance-based method is recommended when the DG ratio cannot be obtained under a wide-angle measurement (for example, in winter, the range of angle variation is small) or when the atmospheric condition is unstable.

The vicarious absolute radiometric calibration coefficients of SDGSAT-1 MII obtained based on this experiment have been provided to users at http://www.sdgsat.ac.cn/, (accessed on 9 June 2022). This research needs to be extended to other radiometric calibration sites, especially, the Radiometric Calibration Network (RadCalNet). Furthermore, extensive evaluation and validation campaigns should be performed to monitor the long-term radiometric performance of SDGSAT-1 MII.

**Author Contributions:** Conceptualization, H.Z and Z.C.; investigation, Z.C., Y.H. and L.Y.; data curation, Z.C.; formal analysis, H.Z., C.D. and C.M.; validation, Z.C.; writing—original draft preparation, Z.C.; writing—review and editing, H.Z. and X.-M.L.; project administration, H.Z., X.-M.L. and C.M. All authors have read and agreed to the published version of the manuscript.

**Funding:** This research was funded by the CAS Strategic Priority Research Program (grant number XDA19010402) and the National Natural Science Foundation of China (grant number 41771397, U21A20108).

**Data Availability Statement:** Not applicable.

**Acknowledgments:** The authors thank the personnel at the Dunhuang National Climate Station for providing the AOD, water vapor content, and radiosonde data, as well as Ning Wang, Yongzheng Ren, and Hujun Liang for their help with the ground reflectance measurements.

**Conflicts of Interest:** The authors declare no conflict of interest.

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
