# Peer review of "Vicarious Radiometric Calibration of the Multispectral Imager Onboard SDGSAT-1 over the Dunhuang Calibration Site, China"

_remotesensing, doi:10.3390/rs15102578_

Round 1

Reviewer 1 Report

This study provides pretty comprehensive description of an in-situ vicarious calibration assessment for the SDGSAT-1 MII sensor using data collected at the Dunhuang calibration site. I only have a few inputs for the manuscript:

1) It is necessary to provide details about any onboard calibration components used in the SDGSAT-1 MII sensor and any anticipated degradation impact, particularly on the blue spectral region. If there is no onboard calibrators, will the sensor completely rely on the vicarious calibration?

2) Pg 21, for statement "relative differences between the simulated results under different angles and the actual observation results, calculated using Equation (14), were regarded as the calibration uncertainty caused by a variation in the viewing geometry", my concern is if any BRDF impact of the underlying surface, i.e., the test site, in Eq (14) have been considered based on actual measurements over the site or only use the angular matched surface and TOA overpasses in order to reduce the BRDF impact? 

Author Response

Dear Reviewer,

Thanks very much for taking your time to review this manuscript. I really appreciated all your comments and suggestions. Please find my itemized responses and my revised version of the manuscript in the attached files.

Thanks again!

Reviewer 2 Report

The authors give a comprehensive and detailed introduction of on-orbit in-situ absolute radiometric calibration and uncertainty analysis of Multispectral Imager onboard SDGSAT-1 satellite. Study background, aim, major findings, and conclusions were presented. And the results showed the reasonable calibration accuracy by using the reflectance-, irradiance- and improved irradiance-based methods. This work is of great significance to the calibration of SDGSAT-1 MII data. Overall, the topic is suitable for Remote Sensing. However, some concerns should be addressed further to make the work being worthy of publication. And it is recommended to accept after revision.

The detailed suggestions are as follows:

1. The ps in Equation (11) is not defined, please check and revise.

2. Are “radiance” and “transmittance” countable nouns? Please carefully check and confirm throughout the whole manuscript.

3. The formats of Table 9, Table 10, and Table 14 are different from that of other Tables, please check and revise.

4. Please check and revise the expressions like “uncertainty error” in a consistent style throughout the whole manuscript.

5. In the “Discussion” section, you clearly presented the data in figures and tables with certain discussions, but the discussion of the data presented in the “Results” section is not enough. Please supplement the analysis of the section (3) and section (4) in the “Results” section.

6. In the “conclusion” section, I suggest you consider adding an overall conclusion on the usability of the SDGSAT-1 data, like you did in the abstract.

Author Response

(The authors gave the same response as above.)

Reviewer 3 Report

The authors described the in-situ vicarious radiometric calibration experiment of the SDGSAT-1 MII at the Dunhuang calibration site. I have found the manuscript interesting as a technical report. However, the following comments need to be addressed to improve the quality of the manuscript.

The manuscript needs extensive English polishing.

The authors didn't discuss their results. Therefore, I am suggesting improving their discussion.

The authors should provide details about the calibration component onboard SDGSAT-1 MII and any anticipated degradation impact. 

The conclusion can be improved, and the overall decision needs to be added.  

Author Response

(The authors gave the same response as above.)

Round 2

Reviewer 2 Report

This manuscript has been modified very well and can be accepted.

Reviewer 3 Report

The authors addressed all raised issues and I can recommend the manuscript for publishing